# Development and Assessment of 1,5–Diarylpyrazole/Oxime Hybrids Targeting EGFR and JNK–2 as Antiproliferative Agents: A Comprehensive Study through Synthesis, Molecular Docking, and Evaluation

**DOI:** 10.3390/molecules28186521

**Published:** 2023-09-08

**Authors:** Kamal S. Abdelrahman, Heba A. Hassan, Salah A. Abdel-Aziz, Adel A. Marzouk, Raef Shams, Keima Osawa, Mohamed Abdel-Aziz, Hiroyuki Konno

**Affiliations:** 1Department of Pharmaceutical Chemistry, Faculty of Pharmacy, Al-Azhar University, Assiut Branch, Assiut 71524, Egypt; salah72aa@yahoo.com (S.A.A.-A.); adel_marzouk77@yahoo.com (A.A.M.); 2Department of Medicinal Chemistry Faculty of Pharmacy, Minia University, Minia 61519, Egypt; heba.hasan@mu.edu.eg (H.A.H.); abulnil@mu.edu.eg (M.A.-A.); 3Department of Pharmaceutical Chemistry, Faculty of Pharmacy, Deraya University, Minia 61768, Egypt; 4National Center for Natural Products Research, School of Pharmacy, University of Missippi, Oxford, MS 38677, USA; 5Emergent Bioengineering Materials Research Team, RIKEN Centre for Emergent Matter Science, RIKEN, Wako 351-0198, Saitama, Japan; raefshams88@gmail.com; 6Graduate School of Science and Engineering, Yamagata University, Yonezawa 992-8510, Yamagata, Japan; green.ranger.ok@gmail.com

**Keywords:** pyrazole, EGFR, JNK-2, docking simulation, anti-proliferative

## Abstract

New 1,5-diarylpyrazole oxime hybrid derivatives (scaffolds **A** and **B**) were designed, synthesized, and then their purity was verified using a variety of spectroscopic methods. A panel of five cancer cell lines known to express EGFR and JNK-2, including human colorectal adenocarcinoma cell line DLD-1, human cervical cancer cell line Hela, human leukemia cell line K562, human pancreatic cell line SUIT-2, and human hepatocellular carcinoma cell line HepG2, were used to biologically evaluate for their in vitro cytotoxicity for all the synthesized compounds **7a**–**j**, **8a**–**j**, **9a**–**c**, and **10a**–**c**. The oxime containing compounds 8a–j and 10a–c were more active as antiproliferative agents than their non-oxime congeners 7a–j and 9a–c. Compounds **8d**, **8g**, **8i**, and **10c** inhibited EGFR with IC_50_ values ranging from 8 to 21 µM when compared with sorafenib. Compound **8i** inhibited JNK-2 as effectively as sorafenib, with an IC_50_ of 1.0 µM. Furthermore, compound **8g** showed cell cycle arrest at the G2/M phase in the cell cycle analysis of the Hela cell line, whereas compound **8i** showed combined S phase and G2 phase arrest. According to docking studies, oxime hybrid compounds **8d**, **8g**, **8i**, and **10c** exhibited binding free energies ranging from −12.98 to 32.30 kcal/mol at the EGFR binding site whereas compounds **8d** and **8i** had binding free energies ranging from −9.16 to −12.00 kcal/mol at the JNK-2 binding site.

## 1. Introduction

Cancer is the second-leading cause of death worldwide. As a result, the incidence rate of cancer mortality is becoming increasingly significant on a global basis [1,2]. Chemotherapy is one of the cancer treatment options that employs medications that target cell division and angiogenesis or that trigger cancer cell death via multiple signaling pathways. However, due to the adverse effects of chemotherapy and the development of drug resistance in cancer cells, there is an urgent need for the design, synthesis, and development of effective and safe chemotherapy [3,4].

The tyrosine kinase receptor EGFR is crucial for cellular signaling processes such as cell growth, division, differentiation, metabolism, adhesion, and death [5]. The HER family comprises four tyrosine kinase-related receptors (EGFR, HER2, HER3, and HER4). The deregulation of HER family signaling promotes cancer cell survival and proliferation, invasion, metastasis, and angiogenesis [6]. EGFR receptors are overexpressed in a variety of human tumors, including leukemia, breast, ovarian, prostate, colon, renal, pancreatic, and hepatocellular carcinoma [6,7,8,9,10,11,12,13]. Consequently, EGFR inhibition is now recognized as one of the most effective cancer-treatment strategies. Several small molecules that target EGFR are currently accessible clinically, including gefitinib, erlotinib, lapatinib, and dacomitinib [2,4,14].

JNK-2 is a member of the MAP kinase family involved in signaling pathways that has been implicated in several diseases like cancer and inflammatory diseases [15]. Due to the important key roles of JNK-2 in cancer progression through the control of proliferation, differentiation, survival, and migration, JNK-2 becomes an appealing oncogenic target for cancer therapy due to its high expression in a variety of cancers, including colorectal adenocarcinoma, cervical cancer, pancreatic cancer, hepatocellular carcinoma, and leukemia [16,17,18,19,20,21,22]. JNK signaling is apparently involved in cancer development and progression in lymphoma cancer cells through protecting it from apoptosis by decreasing ROS accumulation. Also, JNK regulates micro-RNA-92a and glucose regulating protein 78 (GRP78) in human pancreatic cancer, which promotes cell proliferation and survival. In hepatocellular carcinoma (HCC), the JNK pathway is responsible for its development and progression and becomes the target for the therapeutic treatment of HCC. Otherwise, the blocking of the JNK pathway leads to the inhibition of proliferation human B lymphoma cells due to the downregulation of early growth response gen-1 (Egr-1) protein. Moreover, there is a relation between the JNK pathway and other pathways like kappa B (NF-KB) and p38, which are acting together for the regulation of cell proliferation and survival. Also, there is a close relation between JNK and immune evasion regulatory factors such as transforming growth factor-β (TGF-β) and interferon-γ (IFN-γ) mediate cell survival. In addition, JNK can promote cancer cell survival through autophagy to counteract apoptosis. To date, the majority of JNK inhibitors target the highly conserved ATP-binding site. While a number of these inhibitors were proven in vivo in animal models, they were not applied therapeutically until now due to the lack of their selectivity and side effects. In addition, an increase in the concentration of ATP decreases its efficacy [23].

Pyrazoles are a significant class of heterocyclic chemicals that have a wide range of biological effects, such as anticancer effects [24] and anti-inflammatory [25], antimicrobial [26], antiviral [27], and antitubercular activities [28]. Some pyrazole-containing compounds, such as compounds **I** and **II**, exhibited antiproliferative activity against the human cervical cancer cell line Hela by inhibiting cell migration and potent EGFR tyrosine kinase inhibitory activity with IC_50_ values of 0.07 and 0.06 µM, respectively, in comparison with the positive control erlotinib (IC_50_ = 0.03 µM) [4,29]. Moreover, the selective COX-2 inhibitor compound SC-236 showed antitumor activity through the blocking of tumor promotor-induced activator protein-1 (AP1) activation as a result of the suppression of JNK expression and was used to treat hepatocellular cancer in conjunction with doxorubicin [30]. Furthermore, compound SC74102 displayed JNK-2 inhibitory activity with IC_50_ of 1.35 µ mol/L as well as its p38α inhibitory activity. Additionally, diarypyrazoles have been reported to have STAT3 inhibitory activity, as in compound MNS1-Leu [31], and heat shock protein inhibitory activity, as in compound CCT072453 [32] (Figure 1).

Oximes are the focus of keen interest in medicinal chemistry. The oxime moiety can hydrogen bond to amino acid residues in the active site of many enzymes and is easily coordinated with metal ions. As a result, the oxime moiety can boost the overall binding of the molecule to its binding site. Oximes can also produce nitric oxide free radicals, which have both cytostatic and cytotoxic effects on cancer cells. They can stop cancer cells from spreading and assist macrophages in killing cancer cells. Several targets for combining NO with cancer therapy have been identified, including either the synergistic action of anticancer medications and nitric oxide, enhancing the flow of anticancer therapy by NO to intracellular compartments, or increasing the efficiency of cytostatic therapy and overcoming resistance to anticancer agents [33,34,35]. On the other hand, the introduction of an oxime group into an appropriate chemical backbone is a reasonable approach for the preparation of cytotoxic agents, and many oxime derivatives have been reported to have therapeutic activity for cancer [33,36,37,38]. Also, the introduction of the oxime moiety in some natural compounds such as psammaplin A was responsible for high anticancer activity [39]. Triterpene-derived acylated oximes have demonstrated cytotoxic or antiproliferative action against numerous cancer cell lines [40] and several indirubin oximes showed greater anticancer activity than natural alkaloid indirubin. Furthermore, the oxime derivative of natural alkaloid tryptanthrin showed JNK1/2/3 inhibition [41]. Finally. oximes have been employed in the development of several kinase inhibitors, including those for JNK [41,42], phosphorylase kinase (PhK), and phosphatidyl inositol 3-kinase (PI3K) [43]. Indirubin oximes, for example, have a high affinity for binding to the ATP-binding site of protein kinases involved in the development of tumors, such as cyclin-dependent kinases (CDK), glycogen synthase kinase 3 (GSK) 1, vascular endothelial growth factor receptor 2 (VEGFR-2), c-Src, and casein kinase 2 (CK2). Many of these kinases could serve as molecular targets for drugs that combat cancer (Figure 2) [44].

Based on the information presented above and in a continuation of our efforts to identify small molecules with potential anticancer activity, the goal of this work was to create a hybrid series of 1,5-diarylpyrazole derivatives (Scaffold **A** and **B**) that target EGFR and JNK-2 and contain oxime as a NO release moiety to enhance anticancer activity. Scaffolds **A** and **B** were created to possess the critical pharmacophoric properties of EGFR/JNK-2 inhibitors by employing the ester (Scaffold **A**) or amide moiety (Scaffold **B**), as well as the oxime moiety, to produce hydrogen bonding connections. Furthermore, vicinal 1,5-diarylpyrazole appears to be more adaptable when it comes to accessing both enzymes allosteric hydrophobic regions. The enhancement of the active site flexibility of protein binding can be achieved by employing the sandwiching effect between non-polar amino acid residues of EGFR/JNK-2 binding pockets and aryl moieties of synthesized compounds. Moreover, the presence of a carbonyl group and oxime moiety in the design allows for the establishment of hydrogen bonds with the amino acid residues located in the EGFR/JNK-2 binding pockets. To investigate its SAR, different substitutions (electron donating and withdrawing groups) were applied. Furthermore, distinguishing between scaffolds **A** and **B** ensures that the optimum pharmacophore with the best replacement for enzyme binding is kept. This hybridization was performed to provide a synergistic effect, boost anticancer effectiveness, and/or reduce any adverse effects (Figure 3).

## 2. Results and Discussion

### 2.1. Chemistry

The Claisen condensation of different substituted acetophenone **1a**–**e** with diethyl oxalate in the presence of sodium ethoxide provides 1,3-dicarbonyl compounds (β-diketoester) **2a**–**e** in a good yield. 4-Hydrazinylbenzenesulfonamide hydrochloride **4b** was synthesized through the diazotization of sulfanilamide with sodium nitrite and hydrochloric acid followed by reduction with sodium sulfite in the presence of sodium hydroxide and hydrochloric acid [45]. Diarylpyrazole carboxylate derivatives **5a**–**j** were synthesized through the condensation of 1,3-dicarbonyl compounds (β-diketoester) **3a**–**e** with phenyl hydrazine **4a** directly or in the presence of sodium acetate, as in compound **4b**. Hydrolysis of 1,5-diarylpyrazole ester **5a**–**j** derivatives with alcoholic potassium hydroxide yielded 1,5-diarylpyrazole carboxylic acid derivatives **6a**–**j** [46]. Compounds **7a**–**j** were synthesized according to Steglich esterification through the coupling of 1,5-diarylpyarzole carboxylic acid derivatives **6a**–**j** with 4-hydroxy-3-methoxy acetophenone using EDC as a coupling agent and HOBt as additives in the presence of DIPEA. The structures of the synthesized compounds **7a**–**j** were confirmed through IR, ^1^H-NMR, ^13^C-NMR, and HRMS (ESI) spectroscopy. The IR spectra showed significant stretching bands at 1658–1746 cm^−1^ related to the carbonyl of ester group (COO-Ph) and at 1162–1164 cm^−1^ for compounds **7f**–**j** related to the (SO_2_NH_2_) group. The ^1^H-NMR spectra of compounds **7a**–**j** showed two common singlet peaks at δ 3.71–3.85 ppm related to the methoxy group of acetophenone and at δ 2.51–2.75 ppm attributed to (CO-CH_3_). The ^13^C-NMR spectra of compounds **7a**–**j** showed significant signals related to carbonyl carbon of ketone (CO-CH_3_) which appeared at δ 197.1–197.8 ppm. The peak at δ 160.0–167.8 ppm was related to carbonyl carbon of ester (COO-Ph), at δ 54.96–57.14 ppm, attributed to carbon of the methoxy group that attached to the acetophenone moiety and, at δ 26.7–28.9 ppm, attributed to carbon of the methyl group (CO-CH_3_). The HRMS (ESI) data for compounds **7a**–**j** further confirmed their assigned structure. The *m*/*z* value of the molecular ion peak [M+H]^+^ or [M+Na]^+^ were close to the calculated ones for all target compounds. The target oxime derivatives **8a**–**j** were prepared by refluxing a mixture of ketone intermediates **7a**–**j** and hydroxylamine hydrochloride in absolute ethanol. The chemical structure of the prepared compounds was elucidated using IR, ^1^H-NMR, ^13^C-NMR, and HRMS spectroscopy. The IR spectra of compounds **8a**–**j** were characterized using the appearance of intense broad bands at 3139–3681 cm^−1^ related to the OH group, in addition to (COO-Ph), which exhibited stretching vibration at 1717–1756 cm^−1^. A characteristic feature of the ^1^H-NMR spectra for oximes **8a**–**j** is the appearance of downfield singlets in the range δ 10.36–11.34 ppm related to the hydroxyl group. The resonances of CH_3_ protons were observed in the expected regions at δ 1.90–2.28 ppm and appeared to be more upfield shifted than the CH_3_ protons of the corresponding ketones by 0.40–0.60 ppm due to the low electronegativity of the nitrogen atoms of the oxime relative to the oxygen atoms of the ketone. Also, all the aromatic protons appeared in the expected chemical shift range. One of the characteristic features of ^13^C-NMR spectra of compounds **8a**–**j** is the disappearance of ketonic carbonyl due to their conversation to the ketoxime group (C=N-OH), which appeared at δ 150.8–160.0 ppm. Also, the methyl group attached to ketoxime appeared at δ 11.65–14.71 ppm. HRMS (ESI) data for compounds **8a**–**j** further confirmed their assigned structure. The *m*/*z* value of the molecular ion peak [M+H]^+^ or [M+Na]^+^ were close to the calculated ones for all cases (Figure 1).

Compounds **9a**–**c** were synthesized by activating 1,5-diarypyrazole carboxylic acid derivatives **6f**, **6h**, and **6j** with thionyl chloride in benzene to obtain acyl chloride derivatives, which were coupled with 4-aminoacetophenone by heating in dry DMF in the presence of triethylamine as the base. The structure of the synthesized compounds **9a**–**c** was confirmed using IR, ^1^H-NMR, ^13^C-NMR, and HRMS (ESI) spectroscopy. The IR spectra showed significant stretching bands at 1669–1681 cm^−1^ assigned to (CONH) and at 1160–1161 cm^−1^ related to (SO_2_NH_2_). In the ^1^H-NMR spectra for compounds **9a**–**c**, two singlet peaks were common and appeared at δ 10.53–11.18 ppm related to the amidic (NH) proton and at δ 2.47–2.53 ppm related to (CO-CH_3_). The ^13^C-NMR spectra of compounds **9a**–**c** showed significant signals related to carbonyl carbon of ketone (CO-CH_3_) which appeared at δ 196.0–197.1 ppm, and peaked at δ 164.1–167.9 ppm related to carbonyl carbon of amide (CONH) and at δ 26.6–27.0 ppm related to carbon of methyl (CO-CH_3_). HRMS (ESI) data for compounds **9a**–**c** confirmed their assigned structure. The *m*/*z* value of molecular ion peak [M-1]^−^ or [M+Na]^+^ were close to the calculated ones for all cases. Oxime derivatives **10a**–**c** were synthesized by refluxing a mixture of ketone intermediates **9a**–**c** and hydroxylamine hydrochloride in absolute ethanol. The chemical structure of the prepared compounds was elucidated using IR, ^1^H-NMR, ^13^C-NMR, and HRMS spectroscopy. The IR spectra of compounds **10a**–**c** was characterized by the appearance of intense broad bands at 3220–3681 cm^−1^ related to the OH and NH groups, in addition to the (CO-NH) and (SO_2_NH_2_) groups that exhibited stretching vibration at 1677–1680 cm^−1^ and 1161–1162 cm^−1^, respectively. A characteristic feature of the ^1^H-NMR spectra for oximes **10a**–**c** was the appearance of downfield singlets in the range δ 8.75–10.9 ppm, related to the hydroxyl group. The resonances of NH and CH_3_ protons were observed in the expected regions at δ 10.30–10.36 ppm and δ 2.06–2.08 ppm, respectively. The CH_3_ protons appeared to be more upfield shifted than the CH_3_ protons of the corresponding ketones by 0.40–0.45 ppm due to the low electronegativity of the N atom of the oxime relative to O atom of the ketones. One of the characteristic features of the ^13^C-NMR spectra of compounds **10a**–**c** is the disappearance of ketonic carbonyl due to its conversation to the ketoxime group (C=N-OH), which appeared at δ 153.2–159.2 ppm. Also, the methyl group attached to ketoxime appeared at δ 12.1–17.3 ppm. The HRMS (ESI) data for compounds **10a**–**c** further confirmed their assigned structure. The *m*/*z* value of the molecular ion peak [M-H]^−^ or [M+H]^+^ were close to the calculated ones for all target compounds (Figure 2).

### 2.2. Biology

#### 2.2.1. In Vitro Antiproliferative Screening Activities

Compounds **7a**–**j**, **8a**–**j**, **9a**–**c**, and **10a**–**c** were evaluated for in vitro anticancer activities against five different cancer cell lines, namely, human colorectal adenocarcinoma cell lines DLD-1, human cervical cancer cell line Hela, human pancreatic cancer cell line SUIT-2, human myelogenous leukemia cell line K562, and human hepatocellular carcinoma cell line HepG2 using WST-8 assay at concentrations of 100 µM to investigate the growth inhibition percent (GI%) of each compound using daunorubicin as a reference drug [47]. From the screening results in Table 1, compounds **7a**–**j** and **9a**–**c**, which are 1,5-diarylpyrazole acetophenone derivatives, displayed considerable cytotoxicity towards the pancreatic cell line SUIT-2 with GI% ranging from 68 to 104% for compounds **7a**–**j** and 42 to 60% for compounds **9a**–**c**. Compounds **7b** (R_1_ = CH_3_, R_2_, R_3_ = H) demonstrated moderate to high cytotoxicity against five cancer cell lines with GI% ranging from 46 to 103%, while compound **7d** (R_1_ = Cl, R_2_, R_3_ = H) established an excellent cytotoxicity towards DLD-1, Hela, and SUIT-2 with a GI% of 72%, 104%, and 104%, respectively, and moderate cytotoxicity against K562 cell line with a GI% 50. Furthermore, compound **7i** (R_1_ = Cl, R_2_ = H, R_3_ = SO_2_NH_2_) displayed superior antiproliferative activity against Hela, K562, and SUIT-2 cell lines with a GI% of 101%,101%, and 101% and moderate activity against HepG2 cell line with a GI% of 63%. Importantly, all oxime derivatives **8a**–**j** and **10a**–**c** exhibited marked antiproliferative activity in comparison with ketone derivatives **9a**–**c** and **11a**–**c** as a result of the role of oxime moiety in cytotoxicity. Compounds **8b**–**j** demonstrated a significant cytotoxicity against SUIT-2 and HepG2 cell lines with a GI% ranging from 51 to 107%. Compounds **8b** (R_1_ = CH_3_, R_2_, R_3_ = H), **8f** (R_1_, R_2_ = H, R_3_ = SO_2_NH_2_), and **8g** (R_1_ = CH_3_, R_2_ = H, R_3_ = SO_2_NH_2_) exhibited broadness cytotoxicity in five cancer cell lines with a GI% ranging from 60 to 107%. Moreover, compounds **8a**–**i** exhibited high antiproliferative activity against leukemia cell line K562 with a GI% ranging from 67 to 99%. Compound **8e** (R_1_, R_2_ = OCH_3_, R_3_ = H) displayed remarked cytotoxicity in Hela, K562, SUIT-2, and HepG2 with a GI% of 77%, 93%, 83%, and 99%, respectively, while compound **8i** (R_1_ = Cl, R_2_ = H, R_3_ = SO_2_NH_2_) exhibited high antiproliferative activity against Hela, K562, SUIT-2, and HepG2 with a GI% of 100%, 97%, 98%, and 90%, respectively. Furthermore, compounds **10a**–**c** showed moderate to high cytotoxicity in pancreatic cancer cell line SUIT-2 with a GI% of 88%, 50%, and 103%, respectively; only compound **10c** (R_1,_ R_2_ = 3,4-OCH_3_) demonstrated antiproliferative activity in the five cancer cell lines with a GI% ranging from 52 to 103%. So, as a conclusion on the SAR study of those compounds as antiproliferative agents, the oxime moiety potentiated the anticancer activity and electron donating groups on R_1_ and R_2_ played an important role on this activity. Moreover, the sulfamoyl moiety on R_3_ seems to play a potential role in the activity of the prepared compounds (Table 1).

#### 2.2.2. In Vitro Cytotoxicity Measurements (IC_50_) against Five Cancer Cell Lines

For further investigation, compounds with committed antiproliferative activity against five cancer cell lines (DLD-1, Hela, K562, SUIT-2, and HepG2) at 100 µM were selected to measure growth inhibition percentage using WST-8 assay at different six concentrations of 1, 10, 20, 50, 80, and 100 µM for calculating their IC_50_ using the daunorubicin as reference. All selected compounds and daunorubicin were recorded as the minimum concentration required to inhibit half cell growth (IC_50_) and the results are listed in Table 2.

The target oxime derivatives exhibited promising antiproliferative activity against five cancer cell lines, as listed in Table 2, more so than the corresponding ketone, such as compounds **8b** (R_1_ = CH_3_, R_2_, R_3_ = H), **8d** (R_1_ = Cl, R_2_, R_3_ = H), **8g** (R_1_ = CH_3_, R_2_ = H, R_3_ = SO_2_NH_2_), **10a** (R_1_, R_2_ = H), and **10b** (R_1_ = OCH_3_, R_2_ = H), which showed remarkable cytotoxicity against the DLD-1 cell line with IC_50_ of 10, 14.4, 32.30, 26, and 36 µM, respectively, in comparison with daunorubicin (IC_50_ = 30 µM). Also, compounds **8g**, **8i** (R_1_ = CH_3_, R_2_ = H, R_2_ = SO_2_NH_2_), and **10c** (R_1_, R_2_ = OCH_3_) demonstrated high antiproliferative activity against the Hela cell line with IC_50_ of 8, 13, and 5 µM, respectively, while compounds **8b**, **8d**, **8e** (R_1_, R_2_ = OCH_3_, R_3_ = H), **8f** (R_1_, R_2_ = H, R_3_ = SO_2_NH_2_), and **8h** (R_1_ = OCH_3_, R_2_ = H, R_3_ = SO_2_NH_2_) showed a moderate antiproliferative activity with IC_50_ of 32, 57, 22, 22, and 74 µM in comparison with daunorubicin. Moreover, compounds **8b**, **8d**, **8f**, **8g**, and **10a** established excellent anticancer activity in comparison with daunorubicin (IC_50_ = 13 µM) against the human myelogenous leukemia cell line K562 with IC_50_ of 13, 9, 15.6, 7.6, and 16 µM. Compounds **8a** (R_1_, R_2_, R_3_ = H), **8e**, **8h**, and **10c** exhibited good anticancer activity at the same cell line with IC_50_ of 22, 20, 21, and 29 µM. Furthermore, compounds **8b**, **8g**, **8h**, and **10c** showed a significant anticancer activity against the human pancreatic cancer cell line SUIT-2 with IC_50_ of 27, 19, 26, and 13 µM, respectively. Compounds **8e** and **8g** demonstrated excellent antiproliferative activity better than daunorubicin (IC_50_ = 22 µM) against the hepatocellular carcinoma cell line HepG2 with IC_50_ of 4.7 and 12.3 µM, respectively. In addition, compounds **8d** and **8f** showed equal anticancer activity to daunorubicin at the same cell line with IC_50_ of 23.3 and 22.3 µM, respectively. Unlike to anticancer activity of ketone derivatives **7a**–**j**, compounds **7b** (R_1_ = CH_3_, R_2_, R_3_ = H) and **7g** (R_1_ = CH_3_, R_2_ = H, R_3_ = SO_2_NH_2_) displayed excellent antiproliferative activity in comparison with daunorubicin against human colorectal adenocarcinoma DLD-1 with IC_50_ of 13 µM. Also, compound **7d** established remarkable anticancer activity towards the human cervical cancer cell line Hela with IC_50_ of 15 µM. Substituents on the terminal phenyl ring of the 1,5-diarylpyrazole part showed a significant effect on the biological profile of anticancer activity. Compounds **8b**, **8f**, **8g**, **8h**, and **8i**, which are considered the most potent anticancer oxime derivatives, showed substituents on R_1_, R_2_ = H, CH_3_, and Cl, but when R_1_, R_2_ = OCH_3_, moderate anticancer activity was observed, as in compound **8h**, which indicated the presence of a lipophilic group (Cl, CH_3_) improving anticancer activity. Also, the sulfamoyl group at the *para* position of the phenyl ring of the diarylpyrazole part is essential for the broadness of anticancer activities such as compounds **8g**, **8i**, **8h**, and compound **10c** as a result of the hydrogen bonding formation on the active site. Meanwhile, compound **10c**, R_1_, R_2_ = OCH_3_ established good anticancer activity with (IC_50_ = 5–29 µM) and the remaining compounds in scaffold B showed weak anticancer activity. The difference between the hydrophilic and hydrophobic substitutions in scaffold A, as well as the presence of the methoxy group in the 4-hydroxy-3-methoxyl acetophenone carrying oxime moiety, led to scaffold A superior anticancer efficacy compared with scaffold B. The anticancer properties of 3,4-di-OCH_3_-containing compounds, however, were good in both scaffolds (Table 2).

#### 2.2.3. Evaluation of EGFR and JNK-2 Inhibitory Activity

Being over-expressed in a variety of human cancers and connected to cancer proliferation, angiogenesis, and metastasis, EGFR has received substantial study and clinical validation as a target for cancer treatment. Most of these medications are designed to bind to the ATP active site of EGFR-TK. The structural study of previously reported instances of tyrosine kinase anticancer medications served as the basis for the introduction of such drugs [4,6]. Herein, the human EGFR-TK Elisa kit assay was performed to evaluate the in vitro EGFR-inhibitory potency of the more active anticancer compounds **8b**, **8d**, **8g**, **8i**, and **10c** using the multi-target kinase inhibitor drug sorafenib as a reference using quantitative sandwich enzyme immunoassay technology [48]. The test investigated the potential of the test compounds to bind to EGFR, resulting in the suppression of epidermal growth factor from binding to EGFR that led to the inhibition of receptor dimerization and tyrosine autophosphorylation and the suppression of cancer cell proliferation. The screening results of the EGFR inhibitory activity of tested compounds and sorafenib expressed as IC_50_ in µM are recorded in Table 3. The results showed that compound **8g** (R_1_ = CH_3_, R_2_ = H, R_3_ = SO_2_NH_2_), **8i** (R_1_ = Cl, R_2_ = H, R_3_ = SO_2_NH_2_), and **10c** (R_1_, R_2_ = OCH_3_) exhibited moderate EGFR inhibitory activity with IC_50_ of 18, 21, and 12 µM, respectively. On the other hand, compound **8d** (R_1_ = Cl, R_2_, R_3_ = H) displayed successful EGFR inhibitory activity with IC_50_ of 8 µM in comparison with the positive control drug sorafenib (IC_50_ = 3.5 µM). These findings revealed that lipophilic substitutions on vicinal 1,5-diarylpyarzole, together with the oxime moiety, as in compound **8d**, have good fitting and binding on EGFR, making it an effective EGFR inhibitor.

A member of the MAP kinase family involved in signaling pathways, JNK-2 has been linked to a number of diseases including cancer and inflammatory diseases [15]. As a consequence, this family receives lots of attention for small molecule therapeutic targeting [4,15]. JNK-2 inhibitors go into one of two categories: the DFG-in conformation (open conformation) is the target of type I inhibitors, whereas the DFG-out conformation (closed conformation) is the target of type II inhibitors [4,15,16]. In light of JNK-2′s crucial function in human malignancies due to its contribution to a number of cancer-related pathways and the reported role of celecoxib as a JNK inhibitor, we examined the JNK-2 inhibitory activity of compounds **8d**, **8g**, and **8i** using the multitarget kinase drug sorafenib as a reference [49,50]. The JNK-2 inhibitory activity of the selected compounds was investigated in vitro using a simple step ELISA kit for the quantitative measurement of JNK-2 (pT138/Y185) protein in human cells, which investigated the possible binding of tested compounds in the ATP binding site of JNK-2, leading to the inhibition of substrate binding on the JNK-2 enzyme and the inhibition of the JNK-2 pathway that could explain the antiproliferative activity of these compounds [51]. Screening results of JNK-2 inhibitory activity of tested compounds and sorafenib expressed as IC_50_ in µM are recorded in Table 3. The results showed that the oxime derivative **8i** is a potent inhibitor of JNK-2 with an IC_50_ of 1 μM, the same as the activity of the multi-target kinase sorafenib. These results indicate that the anticancer activity of compound **8i** was due to the dual inhibition of EGFR and JNK-2. Also, compound **8d** showed moderate JNK-2-inhibiting activity with an IC_50_ of 49 μM, which indicated its anticancer activity because of the dual inhibition of EGFR and JNK-2. It was clear from these data that the oxime moiety and *p*-Cl substitution improved the anticancer activity of two compounds in addition to the sulfamoyl moiety that makes compound **8i** more potent as a JNK-2 inhibitor (Table 3).

#### 2.2.4. Cell Cycle Analysis and Apoptosis Detection

##### Cell Cycle Analysis

This analysis was applied to investigate the effects of tested compounds on cell cycle distribution and on cell-death associated DNA fragmentation. The Hela cell line was analyzed flow cytometrically after propidium iodide (PI) staining, following the treatment of compounds **8g** and **8i**. As shown in Figure 4, compound **8g** established G2/M arrest of Hela cancer cells indicating cell death and DNA fragmentation and the percentage of Hela cells at the G2 phase increased from 7.3% (DMSO treated Hela cell) to 16.5% after 24 h and 20.5% after 48 h. While compound **8i** showed a combined S phase and G2 phase arrest and the percentage of Hela cells at the S phase increased from 8.47% (control) to 59.4%, while the percentage of Hela cells at the G2 phase rose from 7.3% (control) to 18.2% (Figure 4).

##### Apoptosis Assay

For further investigation of the anticancer activity of compounds **8g** and **8i**, studies of apoptotic changes after treatment of Hela cells with these inhibitors were examined using fluorescent microscope and flowcytometry. Moreover, to discriminate between apoptosis and necrosis after the treatment of Hela cell with two inhibitors **8g** and **8i** at different concentrations, a fluorescent microscope was used to distinguish between apoptosis and necrosis based on their characteristic difference in morphology using annexin V and PI [52]. As shown in Figure 5, compounds **8g** and **8i** established remarkable apoptosis at low concentrations, which were marked in green because of the binding of annexin-v with phosphatidylserine. This was exposed in the cell membrane of the Hela cells after treatment with two inhibitors due to plasma membrane sprouting and chromatin concentration. The prolonged incubation of Hela cells with two inhibitors directed cells to necrosis, which was marked in red because of the PI staining of necrotic cells as a result of losing their dye-excluding ability owing to a loss of plasma membrane integrity and the dissolution of nuclear chromatin [53] (Appendix A).

Subsequently, the flowcytometric analysis of the effect of compounds **8g** and **8i** on Hela cell apoptosis at a concentration double the IC_50_ of each compound for 24 h using annexin v/PI was investigated and the apoptotic marker changes for each compound on Hela cells were analyzed in comparison with the control untreatable Hela cells. As apparent in Figure 6 and Figure 7, the parentage of apoptotic Hela cells was increased significantly after treatment with compounds **8g** and **8i** from 1.8% for the control untreatable Hela cells to 34.7% for compound **8g** and 17.3% for compound **8i**. These results demonstrate that those apoptotic cells were increased after treatment of these compounds as result of antiproliferative activity not due to cytotoxicity (Figure 5 and Figure 6).

#### 2.2.5. Evaluation of Cytotoxicity towards the Normal Cell Line PC12

The evaluation of normal cell viability is fundamental in analyzing the efficacy of target compounds and is often used in conjunction with cytotoxicity tests to help understand how target compound toxicity affects normal cell health. To investigate the selectivity of the target compounds towards cancer cells, the cytotoxicity of compounds **8g** and **8i** was measured against the normal cell line PC12 (rat adrenal-derived pheochromocytoma cells) using a WST-8 assay at six concentrations 1, 10, 20, 50, 80, and 100 μM to calculate CC_50_ in comparison with daunorubicin as a reference drug. The results showed that no cytotoxicity was observed with compound **8i** against PC12 cell line with CC_50_ > 100 μM. While compound **8g** showed a moderate cytotoxicity with CC_50_ of 16.1 μM in comparison with daunorubicin, which established high cytotoxicity towards the PC12 cell line with percentage growth inhibition of 113% at 100 μM, 107% at 50 μM, and 81% at 1 μM. These results indicated that these target compounds had no or little cytotoxicity against the normal cell line and demonstrated selectivity towards cancer cells.

### 2.3. Measurement of Nitric Oxide Release

For the indirect determination of NO, the Griess colorimetric approach was utilized, which includes spectrophotometry measurements of the stable decomposition products NO_2_^−^ and NO_3_^−^. This method requires that NO_3_^−^ is first reduced to NO_2_^−^ and then NO_2_^−^ is determined by the Griess reaction that includes two steps, the first step is a diazotization reaction in which the NO-derived nitrogen agent, dinitrogen trioxide (N_2_O_3_), which is produced by the spontaneous oxidation of NO with sulfanilamide to form the diazonium ion, is used. The second step is the coupling of diazonium with *N*-(1-napthyl)ethylenediamine dihydrochloride (NEDD) to form a strongly absorbed azo colorimetric product at λ_max_ 546 nm. In order to evaluate thiol-induced NO generation from the appropriate compounds, including NO-donating oximes **8a**–**b**, **8d**–**i**, and **10a**–**c**, they were incubated in aqueous phosphate buffer of pH 7.4 in the presence of excess *N*-acetylcysteine, which serves as a source of thiols that are essential for the release of NO from oximes [34,54]. The signal intensity of the dye is proportional to the amount of NO released. To quantify the amount of NO released, a standard curve was made by measuring the change in absorbance of various concentrations of standard sodium nitrite solutions treated by the same way [34]. The results expressed as amount of NO released (mol/mol) are listed in Table 4. The obtained results indicated that the NO-donating oximes **8a**–**b**, **8d**–**i**, and **10a**–**c** achieved the maximum amount of NO released after 2 h. The data recorded in Table 4 indicated that the NO-donating oximes **8a**, **8b**, **8d**, **8e**, **8f**, **8g**, **8h**, **8i**, and **10a**–**c** achieved the maximum amount of NO release after 2 h. At the first hour, the amount of nitric oxide released increased in compounds **8i**, **8b**, and **8h**, while the maximum release of nitric oxide occurring in compounds **8b** and **8h** was at 2 h. Then, the amount of nitic oxide released declined, which may explain the biological importance of oxime moiety as a source of nitric oxide release in comparison with ketone intermediates **7a**–**j** and **9a**–**c**, as shown in the anticancer activity of oxime derivatives.

### 2.4. Docking

#### 2.4.1. In Silico Molecular Docking Study into EGFR

For the mechanistic investigation of the antiproliferative activity of compounds **8d**, **8g**, **8i**, and **10c**, in silico simulation studies targeting EGFR tyrosine kinase domain (Appendix A) using sorafenib, a multitarget kinase inhibitor drug used as a reference, were undertaken. For the Epidermal Growth Factor Receptor (EGFR), the structural models of the selected ligands were built against the human EGFR complexed with AZD9291 inhibitor (2.80 Å; PDB ID: 4ZAU) [53]. Since the docking simulation scored the ligands based on the structural compatibility and the electrostatic potential, the ligands differentially bound the active site of EGFR at different affinities, as illustrated in Appendix A. As recorded in Table 3, the predicted binding affinities, in turn, come with the evaluated inhibitory values of the EGFR enzymatic activity (Table 3). Sorafenib and oxime ligands (**8d**, **8g**, **8i**, and **10c**) bound the active site by hydrogen bonding and hydrophobic interactions with respect to sorafenib, which had the highest affinity to the active site. Importantly, the amine group of M793 residue acts as a hydrogen donor that forms hydrogen bonding with the carbonyl groups of the ligands and sorafenib. Like sorafenib, compound **8d** was shown to occupy the active site with a binding affinity higher than the other compounds due to its flexibility to fit the active site (Figure 7A,B and Appendix A). The sandwiching effect was achieved by the hydrophobic contacts formed by the non-polar residues L718, V726, A743, and L792. Compound **10c** showed a relatively similar binding score but without Van der Waals forces (Figure 7E and Appendix A). Compounds **8g** and **8i** showed the same affinity levels to bind the active site. (Figure 7C,D and Appendix A).

#### 2.4.2. In Silico Molecular Docking Study on JNK-2

For further mechanistic investigation of the antiproliferative activity, compounds **8d** and **8i** were investigated for their in silico simulation studies targeting the JNK-2 binding pocket (Appendix A) using sorafenib as reference drug. JNK-2 inhibitors are classified to two types: type I inhibitors target the DFG-in conformation (open conformation), while type II inhibitors target the DFG-out conformation (closed conformation) [4,15,16]. Based on open confirmation, we docked the selected compounds against a pre-defined open conformation of JNK-2 [15]. Sorafenib is a common MAP kinase inhibitor, and it was shown to have a higher binding affinity to the JNK-2 active site compared with the other ligands (Appendix A). Sorafenib extends at the active site from the hinge region (L110 and M111) to the DFG conformation (D169), which makes it structurally compatible to fit the active site. Sorafenib binds to the ATP site of JNK-2 and forms two hydrogen bonds: one with M111 of the hinge and the other with K55 of the *N*-terminal b3-strand. In addition, the binding of sorafenib is supported by Van der Waals forces with the hinge residues E109 and Q117 as well as hydrophobic interactions. Like sorafenib, compound 8i is bound to the active site by the same machinery, forming hydrogen bonds with M111 and K55 (Figure 8A,B and Appendix A). However, the weaker affinity of compound **8i** than that of sorafenib comes from the lesser extent of compound **8i** to the DFG conformation, resulting in weaker hydrophobic interactions. Furthermore, the sulfonamide group of compounds **8i** was shown to be extended outside the binding pocket near the hinge region, describing the group’s lower interactivity. Compound **8i** binding is supported by Van der Waals forces with the hinge region (E109, N114, and Q117). However, the binding energy is lower than that of sorafenib. Compound **8i** binds to the hinge region by only one hydrogen bond with M111 supported by Van der Waals force with Q117 (Figure 8B and Appendix A). On the other hand, compound **8d** hydrophobically binds with the hinge region with extension to the *N*-terminal b3-strand, where it forms a hydrogen bond with K55 and a Van der Waals contact with E109 (Figure 8C and Appendix A). The structural incompatibility of compound **8d** results in a weaker binding score, which in turn, results in weaker binding energy and consequently, a weaker inhibitory effect.

## 3. Conclusions

A series of 1,5-diarylpyrzole derivatives targeting EGFR and JNK-2 were developed and synthesized and biologically evaluated for their anticancer activity against a panel of five cancer cell lines, namely DLD-1, Hela, K562, SUIT-2, and HepG2. Oxime derivative compounds **8a**–**j** and **10a**–**c** showed better anticancer activity than their corresponding ketones. Regarding substituents on the terminal phenyl ring of the 1,5-diarylpyrazole part, they showed a significant effect on the biological profile of anticancer activity especially when R_1_ and R_2_ = H, *p*-CH_3_ and *p*-Cl, such as compounds **8b**, **8f**, **8g**, **8h**, and **8i**, which are considered the most potent anticancer oxime derivatives. Also, oxime derivatives **8g**, **8i**, and **10c** exhibited moderate EGFR inhibitory activity with IC_50_ of 18, 21, and 12 µM respectively, while compound **8d** displayed good EGFR inhibitory activity with IC_50_ of 8 µM. Moreover, compound **8i** showed potent JNK-2 inhibitory activity with IC_50_ = 1.0 µM, similar to the positive reference drug, sorafenib. The selectivity of compounds **8i** and **8g** towards cancer cells rather than normal cells was evaluated and compound **8i** observed no cytotoxicity against the PC12 cell line with CC_50_ > 100 μM, while compound **8g** showed moderate cytotoxicity with CC_50_ of 16.1 μM in comparison with the reference drug daunorubicin. Furthermore, compound **8g** exhibited cell cycle arrest at the G2/M phase in the cell cycle analysis of the Hela cell line, while compound **8i** showed combined S phase and G2 phase arrest. Additionally, Hela cell apoptotic changes after treatments of compounds **8g** and **8i** were investigated using fluorescent microscope and flowcytometry as results of their antiproliferative activity. Lastly, in silico molecular docking studies indicated that compounds **8d**, **8g**, **8i**, and **10c** effectively fit into the EGFR binding site through strong hydrogen bonding and hydrophobic interactions. Specifically, these compounds establish hydrogen bonds by interacting with the amine group of the M793 residue in the ATP binding site of EGFR, through interaction with the carbonyl group and oxime moiety. Furthermore, the hydrophobic components of these compounds interact with the non-polar residues L718, V726, A743, and L792 in the ATP binding site of EGFR. Additionally, compounds **8d** and **8i** also demonstrated favorable binding activity at the ATP site of JNK-**2**. These compounds formed hydrogen bonds with M111 of the hinge and K55 of the N-terminal b3-strand, supported by van der Waals forces involving the hinge residues E109 and Q117 at the ATP site of JNK-2.

## 4. Experimental Section

### 4.1. Chemistry

#### 4.1.1. Material and Equipment

All chemicals used for the preparation of the target compounds are of analytical grade and can be used without further purification. Solvents were purified and freshly distilled before use according to the standard procedures. Reaction progress was monitored using thin layer chromatography (Merck Silica gel 60 F254) on glass plates and visualized with a UV lamp (254 nm). Column chromatography was performed using spherical, neutral silica gel of diameter 40–100 μm (Kanto Chemical Co., Inc., Tokyo, Japan). Melting points were recorded at a ATM-02 (AS ONE, Tokyo, Japan). IR spectra were recorded at FT/IR-Spectrum Two (PerkinElmer, Shelton, CT, USA) at the Faculty of Engineering, Yamagata University, Yonezawa, Japan. ^1^H-NMR (400 or 500 MHz) and ^13^C-NMR (100 or 125 MHz) spectra were recorded on either a JNM-ECX400 or JNM-ECX500 (JEOL, Tokyo, Japan) in Faculty of Engineering, Yamagata University, Yonezawa, Japan. Chemical shifts are reported in ppm relative to tetramethylsilane (0 ppm), chloroform (7.26 ppm: 1H, 77.1 ppm: ^13^C), and dimethyl sulfoxide (2.50 ppm: 1H, 39.6 ppm: ^13^C). Coupling constant (*J*) is measured in hertz (Hz). Multiplicity was designated as follows: s, singlet; d, doublet; t, triplet; q, quartet; p, pentet; dd, doublet of doublet; and m for multiplet. Mass spectra (ESI-MS) were carried out using the AccuTOF JMS-T100LC (JEOL, Tokyo, Japan) at the Faculty of Engineering, Yamagata University, Yonezawa, Japan.

#### 4.1.2. General Procedure for the Synthesis of Ethyl 4-(Substituted Phenyl)-2-Hydroxy-4-Oxobut-2-Enoates (**2a**–**e**)

A mixture of diethyl oxalate (2.92 g, 0.02 mol) and substituted acetophenone derivatives (0.01 mol) in ethanol (50 mL) was added to previously prepared sodium ethoxide (sodium, 0.46 g, 0.02 mol, ethanol 100 mL) at 50 °C. The reaction mixture was heated under reflux for 2–3 h. After cooling, the solvent was removed, and the residue was taken up in water (200 mL) and acidified with concentrated HCl (1 mL). The aqueous mixture was extracted with ethyl acetate (3 × 150 mL). The combined extracts were washed with brine (100 mL), dried (MgSO_4_), and concentrated. The obtained solid was recrystallized from methanol to procure compounds **2a**–**e** and the produced compounds were used in the next step without further purification [2,4,55,56,57].

#### 4.1.3. General Procedure for Synthesis of 4-Hydrazinylbenzenesulfonamide Hydrochloride **4b**

A cold stirred mixture containing sulfanilamide (3.42 g, 0.02 mol), hydrochloric acid (10 mL), and crushed ice (200 g) was diazotized over the course of 30 min by the dropwise addition of sodium nitrite (1.4 g, 0.02 mol) in water (25 mL). With vigorous stirring, the cold diazonium salt solution was quickly added to a well-cooled solution of sodium sulfite (2.52 g) and sodium hydroxide (0.800 g) in water (50 mL). The resulting mixture was subsequently left in an ice bath for 15 min, acidified with 10 mL HCl, and then concentrated. The precipitated 4-hydrazineylbenzenesulfonamide hydrochloride **4b** was recovered and dried.: white crystals; mp: 225 °C (lit. mp: 225 °C); yield 3.9 g (88%) [4,45].

#### 4.1.4. General Procedure for the Synthesis of Ethyl 1,5 Diarypyarzole-3-Carboxylate (**5a**–**j**)

A mixture of diketoesters **2a**–**c** (0.01mole) and phenylhydrazine **4a** (0.01mole) was dissolved in a suitable amount of absolute ethanol (40 mL) and refluxed for 5 h to produce compounds **5a**–**c**. A mixture of diketoesters 2d-f and 4-hydrazinylbenzenesulfonamide hydrochloride **4b** was refluxed in absolute ethanol for 5 h in the presence of sodium acetate (0.02 mole) to produce compounds **5d**–**f**. The content of reaction mixture was evaporated under vacuum and the crude product was purified using column chromatography [2,4,46].

**Ethyl 1,5**–**diphenyl**–**1*H***–**pyrazole**–**3**–**carboxylate (5a):** Reddish brown solid; yield (75%); mp: 85–87 °C (lit. 86 °C) [58].**Ethyl 1**–**phenyl**–**5**–**(p**–**tolyl)**–**1*H***–**pyrazole**–**3**–**carboxylate (5b):** Reddish solid; yield (80%), mp: 87–88 °C (lit. 84–86 °C) [59].**Ethyl 5**–**(4**–**methoxyphenyl)**–**1**–**phenyl**–**1*H***–**pyrazole**–**3**–**carboxylate (5c):** Reddish brown solid; yield (81%); mp: 97–99 °C (lit. 97 °C) [55].**Ethyl 5**–**(4**–**chlorophenyl)**–**1**–**phenyl**–**1*H***–**pyrazole**–**3**–**carboxylate (5d):** Reddish brown solid; yield (89%); mp: 92–94 °C (lit. 95–97 °C) [55].**Ethyl 5**–**(3,4**–**dimethoxyphenyl)**–**1**–**phenyl**–**1*H***–**pyrazole**–**3**–**carboxylate (5e):** Brownish solid, yield (63%); mp: 174–176 °C (lit. 177 °C) [60].**Ethyl 5**–**phenyl**–**1**–**(4**–**sulfamoylphenyl)**–**1*H***–**pyrazole**–**3**–**carboxylate (5f):** Reddish powder; yield (66%), mp: 192–194 °C (lit. 192) [61].**Ethyl 1**–**(4**–**sulfamoylphenyl)**–**5**–**(p**–**tolyl)**–**1*H***–**pyrazole**–**3**–**carboxylate (5g):** Reddish brown; yield (75%), mp: 227–228 °C (lit. 227 °C) [62].**Ethyl 5**–**(4**–**methoxyphenyl)**–**1**–**(4**–**sulfamoylphenyl)**–**1*H***–**pyrazole**–**3**–**carboxylate (5h):** Reddish brown powder; yield (71%), mp: 207–209 °C (lit. 205–207 °C) [63].**Ethyl 5**–**(4**–**chlorophenyl)**–**1**–**(4**–**sulfamoylphenyl)**–**1*H***–**pyrazole**–**3**–**carboxylate (5i):** Reddish brown powder; yield (80%), mp: 107–109 °C (lit. 108 °C) [63].**Ethyl 5**–**(3,4**–**dimethoxyphenyl)**–**1**–**(4**–**sulfamoylphenyl)**–**1*H***–**pyrazole**–**3**–**carboxylate (5j):** Reddish brown powder; yield (64%); mp: 214–215 °C [64].

#### 4.1.5. General Procedure for the Synthesis of 1,5-Diarypyrazole Carboxylic Acids (**6a**–**j**)

A mixture of methanolic solution of compounds **5a**–**j** (4 mmol), potassium hydroxide (KOH, 20%, 10 mL) was stirred at 60 °C for 4 h. After cooling, the mixture solution was poured into water and acidified with hydrochloric acid solution (1 M) to pH = 3. The aqueous mixture was extracted with ethyl acetate (3 × 50 mL) and the aqueous layer was discarded. The combined organic extracts were dried with anhydrous MgSO_4_. The organic solvent was evaporated under vacuum to obtain solid products **6a**–**j** [2,4,65].

**1,5**–**Diphenyl**–**1*H***–**pyrazole**–**3**–**carboxylic acid (6a):** Brown powder; yield (84%); mp: 180–182 °C (lit. 182–183) [56].**1**–**Phenyl**–**5**–**(p**–**tolyl)**–**1*H***–**pyrazole**–**3**–**carboxylic acid (6b):** Reddish powder; yield (87%); mp: 171–172 °C [66].**5**–**(4**–**Methoxyphenyl)**–**1**–**phenyl**–**1*H***–**pyrazole**–**3**–**carboxylic acid (6c):** Reddish brown powder; yield (79%); mp: 192–195 °C (lit. 196–197 °C) [64].**5**–**(4**–**Chlorophenyl)**–**1**–**phenyl**–**1*H***–**pyrazole**–**3**–**carboxylic acid (6d):** Yellowish brown powder; yield (78%); mp: > 300 °C [62].**5**–**(3,4**–**Dimethoxyphenyl)**–**1**–**phenyl**–***1H***–**pyrazole**–**3**–**carboxylic acid (6e):** Brown powder; yield (84%); mp: 213–214 °C; ^1^H-NMR (400 MHz, DMSO-*d*_6_) δ (ppm): 7.94 7.52 (m, 5H, Ar-H), 7.39 (s, 1H, pyrazole-H), 6.94–6.68 (m, 3H, Ar-H), 3.79 (s, 3H, OCH_3_), 3.77 (s, 3H, OCH_3_); ^13^C-NMR (100 MHz, DMSO-*d*_6_) δ (ppm): 163.36, 160.36, 145.81, 144.09, 142.55, 130.71, 128.76, 127.45, 126.47, 125.40, 122.05, 120.40, 115.26, 110.45, 56.23, 56.12; ESI-MS (LR) *m*/*z* [M+H]^+^ for C_18_H_17_N_2_O_4_ calculated: 325.1, found: 325.3.**5**–**Phenyl**–**1**–**(4**–**sulfamoylphenyl)**–**1*H***–**pyrazole**–**3**–**carboxylica acid (6f):** Yellowish brown powder; yield (78%), mp: 184–186 °C (lit. 188 °C) [65].**1**–**(4**–**Sulfamoylphenyl)**–**5**–**(*p***–**tolyl)**–**1*H***–**pyrazole**–**3**–**carboxylic acid (6g):** Reddish brown powder; yield (88%); mp: 194–195 °C [56].**5**–**(4**–**Methoxyphenyl)**–**1**–**(4**–**sulfamoylphenyl)**–**1*H***–**pyrazole**–**3**–**carboxylic acid (6h):** Brownish powder; yield (73%), mp: 197–198 °C [67].**5**–**(4**–**Chlorophenyl)**–**1**–**(4**–**sulfamoylphenyl)**–**1*H***–**pyrazole**–**3**–**carboxylic acid (6i):** Yellowish brown powder; yield (84%); mp: 212–214 °C [68].**5**–**(3,4**–**Dimethoxyphenyl)**–**1**–**(4**–**sulfamoylphenyl)**–**1*H***–**pyrazole**–**3**–**carboxylic acid (6j):** Reddish brown powder; yield (77%); mp: 206–208 °C; ^1^H-NMR (400 MHz, DMSO-*d*_6_) δ (ppm): 10.76 (s, 1H, OH), 7.99 (s, 1H, Ar-H), 7.91 (d, *J*= 8.00 Hz, 2H, Ar-H), 7.79 (d, *J*= 8.00 Hz, 2H, Ar-H), 7.65–7.60 (m, 4H, 2Ar-H, SO_2_NH_2_), 7.43 (s, 1H, pyrazole-H), 3.89 (s, 6H, 2 OCH_3_);^13^C-NMR (100 MHz, DMSO-*d*_6_) δ (ppm): 163.36, 159.98, 147.36, 145.23, 141.21, 131.47, 130.17, 128.82, 126.87, 125.54, 122.87, 119.99, 115.14, 107.53, 56.38, 56.21; ESI-MS (LR) *m*/*z* [M+H]^+^ for C_18_H_18_N_3_O_6_S calculated: 404.1, found: 404.0.

#### 4.1.6. General Procedure for Synthesis of 4-Acetyl-2-Methoxyphenyl 5-(4-Subistituted-phenyl) 1-(4-Substituted-Phenyl)-1H-Pyrazole-3-Carboxylate (**7a**–**j**)

A mixture of pyrazole carboxylic acid derivatives **6a**–**j** (0.001 mol), 1-ethyl-3-(3-dimethylaminopropyl) carbodiimide hydrochloride (EDC) (0.384 g, 0.002 mol), 1-hydroxybenzotriazole (HOBt) (0.306 g, 0.002 mol), were stirred in dry DMF (5 mL) for 30 min, then N,N-diisopropylethylamine (DIPEA) (0.258 g, 0.002 mol) and 4-hydroxy-3-methoxyacetophenone (0.002 mol) were added to the mixture and stirred for 12 h. Then, 20 mL distilled water was added followed by acidification with dil. HCl. Extraction twice with ethyl acetate and purification were performed by using column chromatography with chloroform as eluent for compounds **7a**–**e** and chloroform: methanol 98:2 for compounds **7f**–**j**.

**4**–**Acetyl**–**2**–**methoxyphenyl 1,5**–**diphenyl**–**1*H***–**pyrazole**–**3**–**carboxylate (7a):** Yellowish brown solid; yield (75%); mp: 98–100 °C; IR (ATR) cm^−1^; 1748 (COO-Ph), 1725 (Co-CH_3_), 1574 (C=C); ^1^H-NMR (500 MHz, DMSO-*d*_6_) δ (ppm): 7.65 (d, *J* = 8.50 Hz, 1H, Ar-H), 7.46 (d, *J* = 8.00 Hz, 2H, Ar-H), 7.43–7.39 (m, 3H, Ar-H), 7.38 (s, 1H, Ar-H), 7.34–7.32 (m, 3H, Ar-H), 7.30 (s, 1H, pyrazole-H), 7.28–7.27 (m, 2H, Ar-H), 6.84 (d, *J* = 8.50 Hz, 1H, Ar-H), 3.75 (s, 3H, OCH_3_), 2.57 (s, 3H, CH_3_); ^13^C-NMR (100 MHz, DMSO-*d*_6_) δ (ppm): 197.35, 160.07, 151.86, 147.61, 146.33, 145.04, 143.48, 142.86, 141.18, 139.68, 136.44, 129.91, 129.45, 129.11, 126.55, 123.97, 122.69, 114.98, 111.62, 56.74, 27.17; ESI-MS *m*/*z* [M+Na]^+^ for C_25_H_20_N_2_NaO_4_ calculated: 435.1320, found: 435.1309.**4**–**Acetyl**–**2**–**methoxyphenyl**–**1**–**phenyl 5**–***p***–**tolyl**–**1*H***–**pyrazole**–**3**–**carboxylate (7b):** Yellowish solid; yield (81%); mp: 110–112 °C; IR (ATR) cm^−1^; 1743 (COO-Ph), 1710 (CO-CH_3_), 1575 (C=C); ^1^H-NMR (500 MHz, CDCl_3_) δ (ppm): 7.65 (s, 1H, Ar-H), 7.53–7.50 (m, 6H, Ar-H), 7.36–7.35 (m, 2H, Ar-H), 7.15 (s, 1H, pyrazole-H), 6.92–6.94 (m, 3H, Ar-H), 3.88 (s, 3H, OCH_3_), 2.62 (s, 3H, CH_3_), 2.33 (s, 3H, CH_3_); ^13^C-NMR (100 MHz, CDCl_3_) δ (ppm): 197.40, 159.67, 150.10, 146.64, 143.95, 142.51, 139.46, 138.90, 136.03, 130.16, 129.85, 128.80, 125.35, 124.00, 122.97, 121.94, 111.97, 109.93, 56.48, 26.65, 21.49; ESI-MS *m*/*z* [M+Na]^+^ for C_26_H_22_N_2_NaO_4_ calculated: 449.1477, found: 449.1483.**4**–**Acetyl**–**2**–**methoxyphenyl**–**5**–**(4**–**methoxyphenyl) 1**–**phenyl**–**1*H***–**pyrazole**–**3**–**carboxylate (7c):** Yellowish brown solid; yield (85%); mp: 69–71 °C; IR (ATR) cm^−1^; 1743 (COO-Ph), 1725 (CO-CH_3_), 1577 (C=C); ^1^H-NMR (500 MHz, DMSO-*d*_6_) δ (ppm): 7.64 (d, *J* = 7.00 Hz, 1H, Ar-H), 7.46 (d, *J* = 8.50 Hz, 2H, Ar-H), 7.42 (s, 1H, Ar-H), 7.38 (d, *J* = 7.00 Hz, 1H, Ar-H), 7.34 (d, *J* = 7.50 Hz, 2H, Ar-H), 7.22 (s, 1H, pyrazole-H), 7.17 (d, *J* = 8.50 Hz, 2H, Ar-H), 6.88–6.85 (m, 3H, Ar-H), 3.84 (s, 3H, OCH_3_), 3.79 (s, 3H, OCH_3_), 2.58 (s, 3H, CH_3_); ^13^C-NMR (100 MHz, DMSO-*d*_6_) δ (ppm): 197.73, 159.99, 151.77, 148.69, 145.56, 143.22, 142.20, 139.78, 136.74, 130.95, 130.16, 129.55, 125.76, 124.02, 122.28, 116.09, 115.07, 111.96, 110.36, 57.17, 55.74, 26.65; ESI-MS *m*/*z* [M+Na]^+^ for C_26_H_22_N_2_NaO_5_ calculated: 465.1426, found: 465.1428.**4**–**Acetyl**–**2**–**methoxyphenyl**–**5**–**(4**–**chlorophenyl) 1**–**phenyl**–**1*H***–**pyrazole**–**3**–**carboxylate (7d):** Yellowish brown solid; yield (73%); mp: 85–87 °C; IR (ATR) cm^−1^; 1739 (COO-Ph), 1728 (CO-CH_3_), 1575 (C=C); ^1^H-NMR (400 MHz, CDCl_3_) δ (ppm): 7.60 (s, 1H, Ar-H), 7.56 (d, *J* = 6.00 Hz, 1H, Ar-H), 7.50–7.46 (m, 4H, Ar-H), 7.32–7.35 (m, 2H, Ar-H), 7.17–7.15 (m, 2H, Ar-H), 7.14 (s, 1H, pyrazole-H), 6.90 (d, *J* = 6.50 Hz, 2H, Ar-H), 3.88 (s, 3H, OCH_3_), 2.51 (s, 3H, CH_3_); ^13^C-NMR (100 MHz, CDCl_3_) δ (ppm): 197.11, 160.00, 150.43, 147.01, 146.31, 143.53, 143.23, 138.32, 135.31, 133.65, 129.23, 128.88, 127.17, 126.08, 125.74, 123.01, 113.97, 111.69, 109.80, 56.14, 27.26; ESI-MS *m*/*z* [M+H]^+^ for C_25_H_20_ClN_2_O_4_ calculated: 447.1106, found: 447.1085.**4**–**Acetyl**–**2**–**methoxyphenyl**–**5**–**(3,4**–**dimethoxyphenyl) 1**–**phenyl**–**1*H***–**pyrazole**–**3**–**carboxylate (7e):** Reddish yellow solid; yield (71%); mp: 74–76 °C; IR (ATR) cm^−1^; 1740 (COO-Ph), 1715 (CO-CH_3_), 1589 (C=C); ^1^H-NMR (500 MHz, DMSO-*d*_6_) δ (ppm): 7.65 (s, 1H, Ar-H), 7.63 (s, 1H, Ar-H), 7.56 (d, *J* = 8.50 Hz, 1H, Ar-H), 7.46 (d, *J* = 8.50 Hz, 2H, Ar-H), 7.38–7.41 (m, 3H, Ar-H), 7.30 (s, 1H, pyrazole-H), 6.97 (d, *J* = 8.50 Hz, 1H, Ar-H), 6.85–6.83 (m, 2H, Ar-H), 3.84 (s, 3H, OCH_3_), 3.76 (s, 3H, OCH_3_), 3.71 (s, 3H, OCH_3_), 2.59 (s, 3H, CH_3_); ^13^C-NMR (100 MHz, DMSO-d6) δ (ppm): 197.38, 160.00, 153.54, 151.71, 149.03, 147.99, 147.03, 142.92, 140.14, 136.74, 130.54, 130.34, 129.55, 127.07, 123.73, 122.65, 121.95, 115.07, 112.67, 112.01, 110.57, 57.15, 56.48, 55.44, 26.95; ESI-MS *m*/*z* [M+H]^+^ for C_27_H_25_N_2_O_6_ calculated: 473.1707, found: 473.1714.**4**–**Acetyl**–**2**–**methoxyphenyl**–**5**–**phenyl 1**–**(4**–**sulfamoylphenyl)**–**1*H***–**pyrazole**–**3**–**carboxylate (7f):** Yellowish solid; yield (65%); mp: 69–72 °C; IR (ATR) cm^−1^; 1735 (COO-Ph), 1724 (CO-CH_3_), 1589 (C=C aromatic), 1162 (SO_2_NH_2_); ^1^H-NMR (500 MHz, DMSO-*d*_6_) δ (ppm): 7.91 (s, 1H, Ar-H), 7.87 (d, *J* = 8.50 Hz, 2H, Ar-H), 7.80 (d, *J* = 8.00 Hz, 1H, Ar-H), 7.68–7.64 (m, 3H, Ar-H), 7.55 (d, *J* = 8.50 Hz, 2H), 7.53 (s, 2H, SO_2_NH_2_), 7.41–7.39 (m, 2H, Ar-H), 7.35 (s, 1H, pyrazole-H), 7.31 (d, *J* = 8.00 Hz, 1H, Ar-H), 3.85 (s, 3H, OCH_3_), 2.57 (s, 3H, CH_3_); ^13^C-NMR (100 MHz, DMSO-*d*_6_) δ (ppm): 197.75, 167.77, 159.18, 151.90, 144.65, 144.14, 142.86, 141.18, 135.65, 132.33,129.48, 129.40, 127.04, 125.27, 124.50, 123.58, 122.00, 112.02, 111.12, 56.74, 28.85; ESI-MS *m*/*z* [M+Na]^+^ for C_25_H_21_N_3_ Na O_6_S calculated: 514.1049, found: 514.1044.**4**–**Acetyl**–**2**–**methoxyphenyl**–**1**–**(4**–**sulfamoylphenyl) 5**–***p***–**tolyl**–**1*H***–**pyrazole**–**3**–**carboxylate (7g):** Yellowish solid; yield (78%), mp: 83–85 °C; IR (ATR) cm^−1^; 1746 (COO-Ph), 1718 (CO-CH_3_), 1595 (C=C), 1162 (SO_2_NH_2_); ^1^H-NMR (500 MHz, DMSO-*d*_6_) δ (ppm): 7.95 (d, *J* = 8.50 Hz, 1H, Ar-H), 7.86 (d, *J* = 8.50 Hz, 2H, Ar-H), 7.69–7.63 (m, 2H, Ar-H), 7.55 (d, *J* = 8.50 Hz, 2H, Ar-H), 7.52 (s, 2H, SO_2_NH_2_), 7.39 (d, *J* = 8.50 Hz, 2H, Ar-H), 7.29 (s, 1H, pyrazole-H), 7.19 (d, *J* = 8.50 Hz, 2H, Ar-H), 3.84 (s, 3H, OCH_3_), 2.60 (s, 3H, CH_3_), 2.28 (s, 3H, CH_3_); ^13^C-NMR (100 MHz, DMSO-*d*_6_) δ (ppm): 197.74, 163.64, 151.71, 146.50, 144.25, 143.93, 142.21, 139.63, 130.19, 129.12, 128.82, 128.45, 127.77, 126.14, 125.04, 123.75, 122.31, 112.64, 110.31, 56.44, 27.25, 21.47; ESI-MS *m*/*z* [M+Na]^+^ for C_26_H_23_N_3_NaO_6_S calculated: 528.1205, found: 528.1205.**4**–**Acetyl**–**2**–**methoxyphenyl 5**–**(4**–**methoxyphenyl) 1**–**(4**–**sulfamoylphenyl)**–**1*H***–**pyrazole**–**3**–**carboxylate (7h):** Yellowish solid; yield (55%), mp: 80–83 °C; IR (ATR) cm^−1^; 1750 (COO-Ph), 1720 (CO-CH_3_), 1585 (C=C), 1164 (SO_2_NH_2_); ^1^H-NMR (500 MHz, DMSO-*d*_6_) δ (ppm): 7.86 (d, *J* = 9.00 Hz, 2H, Ar-H), 7.66 (s, 1H, Ar-H), 7.64 (d, *J* = 7.50 Hz, 2H, Ar-H), 7.55 (d, *J* = 9.00 Hz, 2H, Ar-H), 7.51 (s, 2H, SO_2_NH_2_), 7.39 (d, *J* = 8.00 Hz, 1H, Ar-H), 7.13 (s, 1H, pyrazole-H), 7.26–7.23 (m, 3H, Ar-H), 3.84 (s, 3H, OCH_3_), 3.74 (s, 3H, OCH_3_), 2.60 (s, 3H, CH_3_); ^13^C NMR (100 MHz, DMSO-*d*_6_) δ (ppm): 197.35, 160.08, 151.86, 148.90, 144.65, 143.35, 141.58, 136.44, 130.40, 129.61, 127.43, 126.55, 123.98, 123.58, 121.01, 115.47, 113.69, 111.12, 57.53, 55.85, 26.68; ESI-MS *m*/*z* [M+Na]^+^ for C_26_H_23_N_3_ Na O_7_S calculated: 544.1154, found: 544.1153.**4**–**Acetyl**–**2**–**methoxyphenyl**–**5**–**(4**–**chlorophenyl) 1**–**(4**–**sulfamoylphenyl)**–**1*H***–**pyrazole**–**3**–**carboxylate (7i):** Yellowish brown solid; yield (76%); mp: 71–74 °C; IR (ATR) cm^−1^; 1744 (COO-Ph), 1726 (CO-CH_3_), 1591 (C=C), 1162 (SO_2_NH_2_); ^1^H-NMR (500 MHz, DMSO-*d*_6_) δ (ppm):, 7.89 (d, *J* = 8.00 Hz, 2H, Ar-H), 7.86 (s, 1H, Ar-H), 7.80 (d, *J* = 9.00 Hz, 1H, Ar-H), 7.65 (d, *J* = 9.00 Hz, 1H, Ar-H), 7.57 (d, *J* = 10.00 Hz, 2H, Ar-H), 7.52 (s, 2H, SO_2_NH_2_), 7.47 (d, *J* = 8.00 Hz, 2H, Ar-H), 7.38 (s, 1H, pyrazole-H), 7.33 (d, *J* = 10.00 Hz, 2H, Ar-H), 3.84 (s, 3H, OCH_3_), 2.69 (s, 3H, CH_3_); ^13^C-NMR (100 MHz, DMSO-*d*_6_) δ (ppm): 197.80, 162.85, 159.53, 144.43, 143.25, 143.15, 141.69, 136.56, 132.33, 129.46, 127.90, 127.43, 126.56, 126.09, 123.83, 122.31, 120.09, 112.31, 111.90, 57.14, 27.14; ESI-MS *m*/*z* [M-H]^−^ for C_25_H_20_ClN_3_NaO_6_S calculated: 524.0689, found: 524.0688.**4**–**Acetyl**–**2**–**methoxyphenyl**–**5**–**(3,4**–**dimethoxyphenyl) 1**–**(4**–**sulfamoylphenyl)**–**1*H***–**pyrazole**–**3**–**carboxylate (7j):** Yellowish brown solid; yield (52%); mp: 75–78 °C; IR (ATR) cm^−1^; 1749 (COO-Ph), 1727 (CO-CH_3_), 1579 (C=C), 1162 (SO_2_NH_2_); ^1^H-NMR (500 MHz, DMSO-*d*_6_) δ (ppm): 8.05 (d, *J* = 8.50 Hz, 2H, Ar-H), 7.96 (d, *J* = 7.50 Hz, 1H, Ar-H), 7.80 (s, 1H, Ar-H), 7.74 (d, *J* = 8.50 Hz, 2H, Ar-H), 7.69 (s, 2H, SO_2_NH_2_), 7.56 (d, *J* = 7.50 Hz, 1H, Ar-H), 7.50 (s, 1H, Ar-H), 7.10 (d, *J* = 7.50 Hz, 1H, Ar-H), 7.05 (s, 1H, pyrazole-H), 6.95 (d, *J* = 7.50 Hz, 1H, Ar-H), 3.99 (s, 3H, OCH_3_), 3.89 (s, 3H, OCH_3_), 3.75 (s, 3H, OCH_3_), 2.75 (s, 3H, CH_3_); ^13^C-NMR (100 MHz, DMSO-*d*_6_) δ (ppm): 197.75, 163.93, 151.47, 149.29, 148.89, 145.80, 144.68, 143.33, 142.86, 142.08, 136.04, 129.13, 127.41, 126.55, 125.27, 123.58, 122.29, 121.00, 112.90, 111.12, 109.84, 57.13, 56.74, 54.96, 27.17; ESI-MS *m*/*z* [M+Na]^+^ for C_27_H_25_N_3_NaO_8_S calculated: 574.1260, found: 574.1261.

#### 4.1.7. General Procedure for Synthesis of (E)-4-(1-(Hydroxyimino)Ethyl)-2-Methoxyphenyl 5-(4-Substituted Phenyl)-1-(4-Substituted Phenyl)-1H-Pyrazole-3-Carboxylate (**8a**–**j**)

A mixture of the appropriate ketone derivatives **7a**–**j** (0.001 mol) and hydroxylamine hydrochloride (0.138 g, 0.002 mol) in 30 mL of absolute ethanol was heated under reflux for 8–12 h and then left to cool to room temperature. The separated solid was filtered off, washed with 10% ammonia solution, then with distilled water, dried, and crystallized from absolute ethanol, affording the target products **8a**–**j**.

**(*E*)**–**4**–**(1**–**(Hydroxyimino)ethyl)2**–**methoxyphenyl 1,5**–**diphenyl**–**1*H***–**pyrazole**–**3**–**carboxylate (8a):** Yellowish brown solid; yield (67%); mp: 167–170 °C; IR (ATR) cm^−1^; 3191 (OH), 1756 (C=O), 1594 (C=C aromatic); ^1^H-NMR (500 MHz, DMSO-*d*_6_) δ (ppm): 11.30 (s, 1H, OH), 7.47 (s, 1H, Ar-H), 7.41–7.37 (m, 4H, Ar-H), 7.33–7.30 (m, 3H, Ar-H), 7.28–7.23 (m, 3H, Ar-H), 7.20 (s, 1H, pyrazole-H), 7.00 (d, *J* = 8.00 Hz, 1H, Ar-H), 6.78 (d, *J* = 8.00 Hz, 1H, Ar-H), 3.73 (s, 3H, OCH_3_), 2.06 (s, 3H, CH_3_); ^13^C-NMR (100 MHz, DMSO-*d*_6_) δ (ppm): 160.07, 152.75, 148.50, 148.01, 144.05, 142.67, 139.72, 136.71, 132.58, 130.41, 129.62, 128.72, 127.84, 126.55, 123.37, 119.35, 118.92, 116.26, 109.46, 56.24, 12.14; ESI-MS *m*/*z* [M+Na]^+^ for C_25_H_21_N_3_NaO_4_ calculated: 450.1430, found: 450.1423.**(*E*)**–**4**–**(1**–**(Hydroxyimino)ethyl)**–**2**–**methoxyphenyl 1**–**phenyl**–**5**–***p***–**tolyl**–**1*H***–**pyrazole**–**3**–**carboxylate (8b):** Yellowish green powder; yield (75%); mp: 123–125 °C; IR (ATR) cm^−1^; 3138 (OH), 1736 (C=O), 1593 (C=C aromatic); ^1^H-NMR (500 MHz, DMSO-d6) δ (ppm): 10.36 (s, 1H, OH), 7.40–7.35 (m, 3H, Ar-H), 7.29 (s, 1H, Ar-H), 7.15–7.05 (m, 5H, 4 Ar-H, pyrazole-H), 6.96 (d, *J* = 7.50 Hz, 2H, Ar-H), 6.75 (d, *J* = 7.50 Hz, 2H, Ar-H), 3.74 (s, 3H, OCH_3_), 2.21 (s, 3H, Ph-CH_3_), 2.02 (s, 3H, CH_3_); ^13^C-NMR (100 MHz, DMSO-*d*_6_) δ (ppm): 160.73, 153.20, 152.18, 151.13, 147.97, 147.97, 142.01, 139.43, 136.85, 129.81, 129.06, 128.64, 126.07, 124.01, 119.19, 116.09, 115.42, 111.97, 108.89, 56.78, 21.85, 11.83; ESI-MS *m*/*z* [M+H]^+^ for C_26_H_24_N_3_O_4_ calculated: 442.1761, found: 442.1743.**(*E*)**–**4**–**(1**–**(Hydroxyimino)ethyl)**–**2**–**methoxyphenyl 5**–**(4**–**methoxyphenyl) 1**–**phenyl**–**1*H***–**pyrazole**–**3**–**carboxylate (8c):** Yellowish brown powder; yield (75%); mp: 132–135 °C; IR (ATR) cm^−1^; 3400 (OH), 1739 (C=O), 1590 (C=C aromatic); ^1^H-NMR (500 MHz, DMSO-*d*_6_) δ (ppm): 10.43 (s, 1H, OH), 7.47 (d, *J* = 8.50 Hz, 1H, Ar-H), 7.34 (d, *J* = 10.00 Hz, 2H, Ar-H), 7.25 (s, 1H, Ar-H), 7.18 (d, *J* = 10.00 Hz, 2H, Ar-H), 7.11–7.08 (m, 2H, Ar-H), 7.04 (s, 1H, pyrazole-H), 7.95 (d, *J* = 8.50 Hz, 1H, Ar-H), 6.80–6.77 (m, 3H, Ar-H), 3.68 (s, 3H, OCH_3_), 3.62 (s, 3H, OCH_3_), 2.02 (s, 3H, CH_3_); ^13^C NMR (100 MHz, DMSO-*d*_6_) δ (ppm): 162.15, 159.58, 150.58, 147.22, 145.43, 143.76, 142.47, 140.80, 136.88, 130.90, 129.61, 127.83, 126.55, 123.18, 121.16, 120.61, 115.87, 114.57, 109.43, 56.74, 54.96, 14.71; ESI-MS *m*/*z* [M+H]^+^ for C_26_H_24_N_3_O_5_ calculated: 458.1716, found: 458.1715.**(*E*)**–**4**–**(1**–**(Hydroxyimino)ethyl)**–**2**–**methoxyphenyl 5**–**(4**–**chlorophenyl)**–**1**–**phenyl**–**1*H***–**pyrazole**–**3**–**carboxylate (8d):** Yellowish brown powder; yield (65%); mp: 106–109 °C; IR (ATR) cm^−1^; 3228 (OH), 1744 (C=O), 1593 (C=C aromatic); ^1^H-NMR (400 MHz, DMSO-*d*_6_) δ (ppm): 10.90 (s, 1H, OH), 7.60 (d, *J* = 8.40 Hz, 1H, Ar-H), 7.42 (d, *J* = 10.00 Hz, 2H, Ar-H), 7.39 (d, *J* = 10 Hz, 2H, Ar-H), 7.36 (s, 1H, Ar-H), 7.26–7.21 (m, 3H, 2Ar-H, pyrazole-H), 7.00 (d, *J* = 8.00 Hz, 2H, Ar-H), 6.75 (d, *J* = 8.00 Hz, 2H, Ar-H), 3.74 (s, 3H, OCH_3_), 2.07 (s, 3H, CH_3_); ^13^C-NMR (100 MHz, DMSO-*d*_6_) δ (ppm): 160.00, 153.68, 151.87, 147.38, 143.64, 142.89, 139.83, 139.48, 136.84, 134.19, 132.02, 131.01, 129.71, 128.38, 126.38, 123.19, 119.48, 115.55, 109.63, 55.38, 12.45; ESI-MS *m*/*z* [M+H]^+^ for C_25_H_21_ClN_3_O_4_ calculated: 462.1188, found: 462.1184.**(*E*)**–**4**–**(1**–**(Hydroxyimino)ethyl)**–**2**–**methoxyphenyl 5**–**(3,4**–**dimethoxy phenyl)**–**1**–**phenyl**–**1*H***–**pyrazole**–**3**–**carboxylate (8e):** Yellowish white powder; yield (77%); mp: 115–118 °C; IR (ATR) cm^−1^; 3300 (OH), 1740 (C=O), 1593 (C=C aromatic); ^1^H-NMR (500 MHz, DMSO-*d*_6_) δ (ppm): 10.43 (s, 1H, OH), 7.56 (s, 1H, Ar-H), 7.46 (s, 1H, Ar-H), 7.40 (d, *J* = 6.50 Hz, 1H, Ar-H), 7.25–7.19 (m, 4H, Ar-H), 7.01–6.98 (m, 2H, Ar-H), 7.16 (s, 1H, pyrazole-H), 6.79 (d, *J* = 8.50, 2H, Ar-H), 3.73 (s, 3H, OCH_3_), 3.70 (s, 3H, OCH_3_), 3.65 (s, 3H, OCH_3_), 2.06 (s, 3H, CH_3_); ^13^C NMR (100 MHz, DMSO-*d*_6_) δ (ppm): 161.75, 159.58, 153.64, 150.58, 150.19, 149.29, 148.01, 145.05, 144.15, 142.86, 139.50, 130.41, 127.84, 125.76, 121.67, 121.62, 121.41, 119.71, 115.86, 112.90, 109.84, 56.74, 55.85, 54.56, 11.65; ESI-MS *m*/*z* [M+H]^+^ for C_27_H_24_N_2_NaO_6_ calculated: 488.1816, found: 488.1808.**(*E*)**–**4**–**(1**–**(Hydroxyimino)ethyl)**–**2**–**methoxyphenyl 5**–**phenyl**–**1**–**(4**–**sulfamoyl phenyl)**–**1*H***–**pyrazole**–**3**–**carboxylate (8f):** Yellowish powder; yield (53%); mp: 122–124 °C; IR (ATR) cm^−1^; 3681 (OH), 1744 (C=O), 1590 (C=C aromatic), 1165 (SO_2_NH_2_); ^1^H-NMR (500 MHz, DMSO-*d*_6_) δ (ppm): 11.17 (s, 1H, OH), 7.86 (d, *J* = 6.50 Hz, 2H, Ar-H), 7.55 (d, *J* = 6.5 Hz, 2H, Ar-H), 7.51 (s, 2H, SO_2_NH_2_),7.40–7.35 (m, 4H, Ar-H), 7.32–7.30 (m, 3H, Ar-H), 7.24 (s, 1H, Ar-H) 7.19 (s, 1H, pyrazole-H), 3.79 (s, 3H, OCH_3_), 2.06 (s, 3H, CH_3_); ^13^C-NMR (100 MHz, DMSO-*d*_6_) δ (ppm): 165.53, 159.99, 153.18, 151.12, 145.63, 143.53, 142.52, 141.49, 139.80, 136.74, 130.18, 129.18, 127.77, 126.80, 123.77, 123.30, 118.83, 111.65, 109.60, 55.71, 12.56; ESI-MS *m*/*z* [M+Na]^+^for C_25_H_22_N_4_ Na O_6_S calculated: 529.1158, found: 529.1158.**(*E*)**–**4**–**(1**–**(Hydroxyimino)ethyl)**–**2**–**methoxyphenyl 1**–**(4**–**sulfamoylphenyl)**–**5**–***p***–**tolyl**–**1*H***–**pyrazole**–**3**–**carboxylate (8g):** Yellowish white powder; yield (69%); mp: 195–197 °C; IR (ATR) cm^−1^; 3255 (OH), 1740 (C=O), 1594 (C=C aromatic), 1164 (SO_2_NH_2_), ^1^H-NMR (500 MHz, DMSO-*d*_6_) δ (ppm): 11.34 (s, 1H, OH), 7.95 (d, *J* = 7.50, 1H, Ar-H), 7.86 (d, *J* = 7.5 Hz, 2H, Ar-H), 7.68 (d, *J* = 7.50 Hz, 1H, Ar-H), 7.54 (d, *J* = 8.50 Hz, 2H, Ar-H), 7.52 (s, 2H, SO_2_NH_2_), 7.40 (s, 1H, Ar-H), 7.27 (s, 1H, pyrazole-H), 7.24–7.19 (m, 4H, Ar-H), 3.78 (s, 3H, OCH_3_), 2.28 (s, 3H, CH_3_), 2.16 (s, 3H, CH_3_); ^13^C-NMR (100 MHz, DMSO-*d*_6_) δ (ppm): 160.42, 153.19, 151.27, 145.80, 144.61, 142.05, 139.88, 139.41, 136.67, 130.21, 129.17, 127.98, 127.33, 126.51, 126.10, 123.30, 118.84, 111.67, 110.12, 56.45, 21.50, 12.56; ESI-MS *m*/*z* [M+Na]^+^ for C_26_H_24_N_4_NaO_6_S calculated: 543.1314, found: 543.1304.**(*E*)**–**4**–**(1**–**(Hydroxyimino)ethyl)**–**2**–**methoxyphenyl 5**–**(4**–**methoxyphenyl)**–**1**–**(4**–**sulfamoyl phenyl)**–**1*H***–**pyrazole**–**3**–**carboxylate (8h):** Yellowish white powder; yield (49%), mp: 125–127 °C; IR (ATR) cm^−1^; 3400 (OH), 1742 (C=O), 1596 (C=C aromatic), 1165 (SO_2_NH_2_); ^1^H-NMR (500 MHz, DMSO-*d*_6_) δ (ppm): 11.27 (s, 1H, OH), 7.87 (d, *J* = 8.50 Hz, 2H, Ar-H), 7.55 (d, *J* = 9.50 Hz, 2H, Ar-H), 7.51 (s, 2H, SO_2_NH_2_), 7.40 (s, 1H, Ar-H), 7.24–7.20 (m, 4H, Ar-H), 7.19 (s, 1H, pyrazole-H), 6.95 (d, *J* = 9.50 Hz, 2H, Ar-H), 3.79 (s, 3H, OCH_3_), 3.74 (s, 3H, OCH_3_), 2.16 (s, 3H, CH_3_); ^13^C-NMR (100 MHz, DMSO-*d*_6_) δ (ppm): 161.75, 159.18, 153.97, 151.07, 145.93, 144.64, 143.75, 142.47, 139.90, 136.44, 130.41, 129.62, 127.78, 123.18, 122.03, 121.41, 118.82, 115.47, 111.11, 56.74, 55.85, 12.14; ESI-MS *m*/*z* [M+Na]^+^ for C_26_H_24_N_4_NaO_7_S calculated: 559.1263, found: 559.1274.**(*E*)**–**4**–**(1**–**(Hydroxyimino)ethyl)**–**2**–**methoxyphenyl 5**–**(4**–**chlorophenyl)**–**1**–**(4**–**sulfamoyl phenyl)**–**1*H***–**pyrazole**–**3**–**carboxylate (8i):** Yellowish brown powder; yield (67%); mp: 110–112 °C; IR (ATR) cm^−1^; 3371 (OH), 1743 (C=O), 1595 (C=C aromatic), 1161 (SO_2_NH_2_); ^1^H-NMR (500 MHz, DMSO-*d*_6_) δ (ppm): 10.44 (s, 1H, OH), 7.85 (d, *J* = 10.00 Hz, 2H, Ar-H), 7.76 (d, *J* = 6.50 Hz, 1H, Ar-H), 7.61 (s, 1H, Ar-H), 7.49 (s, 2H, SO_2_NH_2_), 7.40–7.33 (m, 4H, 3Ar-H, pyrazole-H), 7.26 (d, *J* = 10.00 Hz, 2H, Ar-H), 7.19 (d, *J* = 10.00 Hz, 2H, Ar-H), 3.71 (s, 3H, OCH_3_), 1.90 (s, 3H, CH_3_); ^13^C-NMR (100 MHz, DMSO-*d*_6_) δ (ppm): 159.57, 150.82, 144.50, 144.38, 143.30, 141.53, 136.59, 134.61, 132.25,, 129.43, 128.25, 127.46, 126.50, 124.90, 123.26, 120.17, 119.66, 118.84, 109.89, 56.38, 11.76; ESI-MS *m*/*z* [M+1]^+^ for C_25_H_22_ClN_4_O_6_S calculated: 541.0943, found: 541.0948.**(*E*)**–**4**–**(1**–**(Hydroxyimino)ethyl)**–**2**–**methoxyphenyl 5**–**(3,4**–**dimethoxyphenyl)**–**1**–**(4**–**sulfamoylphenyl)**–**1*H***–**pyrazole**–**3**–**carboxylate (8j):** Yellowish powder; yield (48%); mp: 129–131 °C; IR (ATR) cm^−1^; 3300 (OH), 1731 (C=O), 1595 (C=C aromatic); 1164 (SO_2_NH_2_); ^1^H-NMR (500 MHz, DMSO-*d*_6_) δ (ppm): 11.26 (s, 1H, OH), 7.87 (d, *J* = 9.00 Hz, 2H, Ar-H), 7.85 (d, *J* = 9.00 Hz, 2H, Ar-H), 7.52 (s, 2H, SO_2_NH_2_), 7.41 (s, 1H, Ar-H), 7.32 (s, 1H, Ar-H), 7.25–7.24 (m, 1H, Ar-H), 6.93 (d, *J* = 8.50 Hz, 1H, Ar-H), 6.88 (s, 1H, pyrazole-H**)**, 6.81–6.75 (m, 2H, Ar-H), 3.79 (s, 3H, OCH_3_), 3.72 (s, 3H, OCH_3_), 3.60 (s, 3H, OCH_3_), 2.17 (s, 3H, CH_3_); ^13^C-NMR (100 MHz, DMSO-*d*_6_) δ (ppm): 167.78, 159.18, 152.75, 149.29, 148.50, 145.44, 144.65, 143.36, 141.58, 140.30, 136.44, 132.58, 129.12, 127.83, 126.15, 122.30, 121.40, 119.33, 112.91, 110.73, 109.84, 56.28, 56.06, 55.89, 11.75; ESI-MS *m*/*z* [M+Na]^+^ for C_27_H_26_N_4_NaO_8_S calculated: 589.1369, found: 589.1380.

#### 4.1.8. General Procedure for Synthesis of N-(4-Acetylphenyl)-5-(4-Subistitutedphenyl)-1-(4-Sulfamoylphenyl)-1H-Pyrazole-3-Carboxamide (**9a**–**c**)

To the suspension of 1,5-diarylpyrazole carboxylic acid derivatives **6f**, **6h**, and **6j** (0.001 mol) in 20 mL of benzene, thionyl chloride (2 mL) was added and heated under reflux for 4 h. Evaporation of the solvent was carried out under vacuum to give a residue of the corresponding acyl chloride that was utilized in the following steps without purification. A mixture of acyl chloride in dry DMF, few drops of triethylamine, and 4-aminoacetophenone (0.270 g, 0.002 mol) were heated under reflux for 8h. Then, 20 mL of cold distilled water was added, followed by acidification with dil. HCl and extraction twice with ethyl acetate. Purification was performed using column chromatography using chloroform: methanol 98:2 as eluent to afford compounds **9a**–**c** [68].

***N***–**(4**–**Acetylphenyl)**–**5**–**phenyl**–**1**–**(4**–**sulfamoylphenyl)**–**1*H***–**pyrazole**–**3**–**carboxamide (9a):** Yellowish brown solid; yield (88%); mp: 81–83 °C; IR (ATR) cm^−1^; 1725 (CO-CH_3_), 1675 (CONH), 1591 (C=C aromatic), 1161 (SO_2_NH_2_); ^1^H-NMR (500 MHz, DMSO-*d*_6_) δ (ppm): 10.53 (s, 1H, NH), 7.96 (d, *J* = 9.00 Hz, 2H, Ar-H), 7.86 (d, *J* = 9.00 Hz, 2H, Ar-H), 7.68–7.62 (m, 3H, Ar-H), 7.57 (d, *J* = 9.00 Hz, 2H, Ar-H), 7.49 (s, 2H, SO_2_NH_2_), 7.40 (d, *J* = 9.00 Hz, 2H, Ar-H), 7.30–7.32 (m, 2H, Ar-H), 7.20 (s, 1H, pyrazole-H), 2.52 (s, 3H, CH_3_); ^13^C-NMR (100 MHz, DMSO-*d*_6_) δ (ppm): 197.09, 167.91, 154.18, 145.63, 144.56, 143.96, 142.53, 133.27, 132.23, 131.21, 129.83, 127.84, 125.96, 125.26, 120.60, 119.04, 109.61, 26.95; ESI-MS *m*/*z* [M+Na]^+^ for C_24_H_20_N_4_ NaO_4_S calculated: 483.1103, found: 483.1109.***N***–**(4**–**Acetylphenyl)**–**5**–**(4**–**methoxyphenyl)**–**1**–**(4**–**sulfamoylphenyl)**–**1*H***–**pyrazole**–**3**–**carboxamide (9b):** Yellowish brown solid; yield (66%); mp: 74–76 °C; IR (ATR) cm^−1^; 1715 (CO-CH_3_),1681 (CONH), 1594 (C=C aromatic), 1160 (SO_2_NH_2_); ^1^H-NMR (500 MHz, DMSO-*d*_6_) δ (ppm): 11.18 (s, 1H, NH), 7.87 (d, *J* = 7.60 Hz, 2H, Ar-H), 7.84 (d, *J* = 7.6.0 Hz, 2H, Ar-H), 7.80 (d, *J* = 9.00 Hz, 2H, Ar-H), 7.62 (d, *J* = 9.00 Hz, 2H, Ar-H), 7.54 (s, 2H, SO_2_NH_2_), 7.33 (s, 1H, pyrazole-H), 6.98 (d, *J* = 7.60 Hz, 2H, Ar-H), 6.96 (d, *J* = 7.60 Hz, 2H, Ar-H), 3.82 (s, 3H, OCH_3_), 2.47 (s, 3H, CH_3_); ^13^C-NMR (100 MHz, DMSO-*d*_6_) δ (ppm): 195.96, 167.20, 154.55, 146.06, 144.59, 142.88, 133.27, 132.59, 131.09, 130.12, 129.18, 126.18, 125.30, 123.46, 120.64, 118.98, 113.06, 55.42, 26.61; ESI-MS *m*/*z* [M-H]^−^ for C_25_H_21_N_4_O_5_S calculated: 489.1238, found: 489.1254.***N***–**(4**–**Acetylphenyl)**–**5**–**(3,4**–**dimethoxyphenyl)**–**1**–**(4**–**sulfamoylphenyl)**–**1*H***–**pyrazole**–**3**–**carboxamide (9c):** Brownish solid; yield (60%); mp: 81–83 °C; IR (ATR) cm^−1^; 1720 (CO-CH_3_), 1669 (CONH), 1590 (C=C aromatic), 1160 (SO_2_NH_2_); ^1^H-NMR (500 MHz, DMSO-*d*_6_) δ (ppm): 10.55 (s, 1H, NH), 7.95 (s, 1H, Ar-H), 7.92 (d, *J* = 9.00 Hz, 2H, Ar-H), 7.84 (d, *J* = 7.50 Hz, 2H, Ar-H), 7.76 (d, *J* = 8.50 Hz, 1H, Ar-H), 7.72 (d, *J* = 9.00 Hz, 2H, Ar-H), 7.62–7.56 (m, 4H, 2 Ar-H, SO_2_NH_2_), 7.37 (s, 1H, pyrazole-H), 7.01 (d, *J* = 8.50 Hz, 1H, Ar-H), 3.82 (s, 3H, OCH_3_), 3.80 (s, 3H, OCH_3_), 2.53 (s, 3H, CH_3_); ^13^C-NMR (100 MHz, DMSO-*d*_6_) δ (ppm): 197.06, 164.12, 153.21, 152.80, 150.42, 148.06, 145.60, 143.23, 142.71, 132.93, 130.56, 129.86, 129.15, 126.10, 124.03, 123.54, 119.19,116.49, 110.95, 56.76, 55.71, 26.62; ESI-MS *m*/*z* [M-H]^−^ for C_26_H_23_N_4_O_6_S calculated: 519.1344, found: 519.1345.

#### 4.1.9. General Procedure for Synthesis of (E)-N-(4-(1-(Hydroxyimino)Ethyl) Phenyl)-5-(4-Subistituted Phenyl)-1-(4-Sulfamoylphenyl)-1H-Pyrazole-3-Carboxamide (**10a**–**c**)

A mixture of the appropriate ketone derivatives **9a**–**c** (0.001 mol) and hydroxylamine hydrochloride (0.138 g, 0.002 mol) in 30 mL of absolute ethanol was heated under reflux for 8–12 h and then left to cool to room temperature. The separated solid was filtered off, washed with 10% ammonia solution, then washed with distilled water, dried, and recrystallized from absolute ethanol to afford the target products **10a**–**c**.

**(*E*)**–***N***–**(4**–**(1**–**(Hydroxyimino)ethyl)phenyl)**–**5**–**phenyl**–**1**–**(4**–**sulfamoylphenyl)**–**1*H***–**pyrazole**–**3**–**carboxamide (10a):** Yellowish powder; yield (55%); mp: 168–170 °C; IR (ATR) cm^−1^; 3681 (OH), 1680 (CONH), 1598 (C=C aromatic), 1162 (SO_2_NH_2_); ^1^H-NMR (500 MHz, DMSO-*d*_6_) δ (ppm): 10.30 (s, 1H, NH), 8.75 (s, 1H, OH), 7.79 (d, *J* = 8.00 Hz, 2H, Ar-H), 7.63 (d, *J* = 8.00 Hz, 2H, Ar-H), 7.50–7.56 (m, 3H, Ar-H), 7.47–7.44 (m, 2H, Ar-H), 7.40 (s, 2H, SO_2_NH_2_), 7.35 (d, *J* = 10.00 Hz, 2H, Ar-H), 7.29 (s, 1H, pyrazole-H), 7.21 (d, *J* = 10.00 Hz, 2H, Ar-H), 2.06 (s, 3H, CH_3_); ^13^C-NMR (100 MHz, DMSO-*d*_6_) δ (ppm): 160.05, 153.24, 151.71, 148.09, 143.55, 141.92, 133.69, 129.88, 129.39, 129.30, 127.44, 126.52, 126.42, 121.34, 120.78, 120.33, 109.58, 12.14; ESI-MS *m*/*z* [M+H]^+^ for C_24_H_22_N_5_O_4_S calculated: 476.1387, found: 476.1397.**(*E*)**–***N***–**(4**–**(1**–**(Hydroxyimino)ethyl)phenyl)**–**5**–**(4**–**methoxyphenyl)**–**1**–**(4**–**sulfamoylphenyl)**–**1*H***–**pyrazole**–**3**–**carboxamide (10b):** Yellowish brown powder; yield (51%); mp: 110–112 °C; IR (ATR) cm^−1^; 3350 (OH), 1677 (CONH), 1596 (C=C aromatic), 1162 (SO_2_NH_2_); ^1^H-NMR (500 MHz, DMSO-*d*_6_) δ (ppm): 10.31 (s, 1H, NH), 10.02 (s, 1H, OH), 7.84 (d, *J* = 8.00 Hz, 2H, Ar-H), 7.78 (d, *J* = 8.50 Hz, 2H, Ar-H), 7.52 (d, *J* = 8.50 Hz, 2H, Ar-H), 7.46 (d, *J* = 9.00 Hz, 2H, Ar-H), 7.41 (s, 2H, SO_2_NH_2_), 7.30 (s, 1H, pyrazole-H), 6.97 (d, *J* = 8.00 Hz, 2H, Ar-H), 6.91 (d, *J* = 9.00 Hz, 2H, Ar-H), 3.80 (s, 3H, CH_3_), 2.07 (s, 3H, CH_3_); ^13^C-NMR (100 MHz, DMSO-*d*_6_) δ (ppm): 167.18, 155.50, 144.08, 142.51, 133.80, 132.35, 131.38, 130.65, 129.47, 127.11, 126.41, 126.09, 125.97, 123.46, 121.51, 114.33, 111.98, 56.12, 17.34; ESI-MS *m*/*z* [M-H]^−^ for C_25_H_22_N_5_O_5_S calculated: 504.1347, found: 504.1377.**(*E*)**–**5**–**(3,4**–**Dimethoxyphenyl)**–***N***–**(4**–**(1**–**(hydroxyimino)ethyl)phenyl)**–**1**–**(4**–**sulfamoyl phenyl)**–**1*H***–**pyrazole**–**3**–**carboxamide (10c):** Brownish powder; yield (44%), mp: 105–107 °C; IR (ATR) cm^−1^; 3220 (OH), 1680 (CONH), 1591 (C=C aromatic), 1161 (SO_2_NH_2_); ^1^H-NMR (500 MHz, DMSO-*d*_6_) δ (ppm): 10.36 (s, 1H, NH), 10.09 (s, 1H, OH), 7.92 (s, 1H, Ar-H), 7.79 (d, *J* = 7.50 Hz, 1H, Ar-H), 7.63 (d, *J* = 8.50 Hz, 2H, Ar-H), 7.52 (d, *J* = 7.50 Hz, 2H, Ar-H), 7.50–7.44 (m, 4H, 2ArH, SO_2_NH_2_), 7.36 (d, *J* = 8.50 Hz, 2H, Ar-H), 7.27 (s, 1H, pyrazole-H), 6.99 (d, *J* = 7.50 Hz, 1H, Ar-H), 3.77 (s, 3H, OCH_3_), 3.74 (s, 3H, OCH_3_), 2.08 (s, 3H, CH_3_); ^13^C-NMR (100 MHz, DMSO-*d*_6_) δ (ppm): 167.78, 159.18, 152.75, 149.29, 148.50, 145.44, 144.65, 143.36, 140.30, 136.44, 132.58, 129.12, 127.83, 126.15, 122.30, 121.40, 119.33, 112.91, 109.84, 56.19, 55.96, 12.14; ESI-MS *m*/*z* [M-H]^−^ for C_26_H_24_N_5_O_6_S calculated: 534.1447, found: 534.1430.

### 4.2. Measurement of Nitric Oxide Release

#### 4.2.1. Materials and Methods

The nitrite calibration curve in addition to the absorbance of the tested compounds were measured on Shimadzu UV-160, UV-Visible spectrophotometer (Shimadzu, Tokyo, Japan). The tested compounds were prepared as solutions in DMF and diluted with the buffer system until a concentration of 100 mM. *N*-Acetyl cysteine solution was prepared in a concentration of 500 mM in methanol. Griess reagent consists of 0.1% *w*/*v* NED solution in water and sulfanilamide solution (1% *w*/*v* of sulfanilamide in 5% *w*/*v* phosphoric acid). Nitrite standard solution (0.1 M sodium nitrite in water) stock solution was prepared, from which a dilute solution of 100 μM nitrite solution was prepared by dilution 1 mL of the stock solution to 1000 mL with phosphate buffer of pH 7.4. From this solution, 4 serials 2-fold dilutions were performed to generate different concentrations of the nitrite solution (100.00, 50.00, 25.00, and 12.50 μM) and these concentrations were used for nitrite calibration curve.

#### 4.2.2. Preparation of Nitrite Standard Curve

Sulfanilamide and NEDD solutions were kept at 25 °C; 100 mL of sulfanilamide solution was added to each dilution of the prepared standard nitrite solution. The mixture was left at 25 °C for 5–10 min protected from light. To this mixture, 100 mL of the NEDD solution was added and the mixture was again left for 5–10 min at 25 °C protected from light. The absorbance of the formed purple color was measured within 30 min at λ_max_ 546 nm, a blank experiment was performed under the same conditions, the procedure was repeated three times for each dilution of the nitrite, and the average absorbance was calculated. A plot of the average absorbance value for each concentration of the nitrite standard solution as a function of “Y” against nitrite concentration as a function of “X” was constructed to generate a standard nitrite calibration curve at pH 7.4.

#### 4.2.3. NO Release Assay

The amount of NO released from the tested compounds **8a**–**b**, **8d**–**I**, and **10a**–**c** was measured using the Griess colorimetric method [69] either in phosphate buffer of pH 7.4 in the presence of *N*-acetyl cysteine, which serves as a source of thiols. The amount of NO released from the tested compounds was measured relative to NO released from standard sodium nitrite solution.

#### 4.2.4. Procedure

Different solutions of the tested compounds **8a**–**b, 8d**–**i**, and **10a**–**c** in DMF were diluted using phosphate buffer of pH 7.4 till a final concentration of 100 mM (test solutions). To 100 mL of different test solutions, 100 mL of *N*-acetyl cysteine solution was added and the obtained solution was kept in an incubator at 37 °C (treated solutions). The solutions were treated similarly to the nitrite standard solution with Griess reagent components; 100 mL of sulfanilamide solution was added to each tube of the treated solution. The mixture was left at 25 °C for 5–10 min protected from light. To this mixture, 100 mL of the NED solution was added and the mixture was again left at 25 °C for 5–10 min protected from light. The absorbance of the formed purple color, if any, was measured within 30 min at λ_max_ 546 nm. A blank experiment was performed under the same conditions, the procedure was repeated three times for each tested compound and the average absorbance was calculated. The corresponding concentration of nitrite was determined through comparison with the nitrite standard calibration curve and the amount of NO released (attributed by the corresponding nitrite concentration) was calculated as percentage of moles of NO released from 1 mole of the tested compounds [34].

### 4.3. Biology

#### 4.3.1. Materials and Methods

Evaluation of anticancer activity was performed according to the standard water-soluble tetrazolium-8 (WST-8) assay at the Faculty of Engineering, Yamagata University, Yonezawa, Japan, using an MTP-310 absorbance microplate reader. The EGFR inhibitory assay was carried out at the Faculty of Engineering, Yamagata University, Yonezawa, Japan, according to the protocol enzyme linked immunosorbent assay (ELISA) kits (Douset sandwich ELISA test, recombinant mouse EGFR). The JNK-2 inhibitory assay was performed at the Faculty of Engineering, Yamagata University, Yonezawa, Japan, according to the protocol of enzyme-linked immunosorbent assay (ELISA), the assay was carried out using the JNK-2 kit (Simple Step ELISA, pT183/Y185, Abcam Company, Cambridge, CB2 0AX, UK). Apoptosis and cell cycle analysis was performed at Faculty of medicine, Yamagata University, Yamagata, Japan, using BD FACS melody. Molecular docking was performed at the Nano Medical Engineering Laboratory, RIKEN Cluster for Pioneering Researchers, RIEKN, Japan, using ICM-Pro 3.8 software (MolSoft L.L.C., San Diego, CA, USA).

#### 4.3.2. Evaluation of Anticancer Activity

According to the standard water-soluble tetrazolium-8 (WST-8) assay, the current synthesized compounds have been tested for their anticancer activities against different five cancer cell lines; DLD-1, Hela, K562, SUIT-2, and HepG2 and daunorubicin was used as reference drug by the WST-8 assay. The five cells were maintained in a suspension culture, (Dulbecco’s modified eagle medium (DMEM) for SUIT-2, Hela, and HepG2 or PRIM for K562 and DLD), supplemented with 5% FBS (Fetal Bovine Serum) containing 1% of a penicillin-streptomycin (1:1) mixture. A 100 μL aliquot of cells (10,000 cells/mL) was added to a 96 well plate and incubated for 24 h at 37 °C in a humidified incubator containing 5% CO_2_ in the air. After 24 h, a 10 μL aliquot of test compound (concentrations varying in the range of 10–150 µM) was added to each of the 96 wells and incubated for 24 h. Then A 10 μL WST-8 solution (mixture of WST-8 and 1-methoxy PMS) was added to each well and the incubation continued for 3 h. The visible absorbance at 450 nm and 630 nm as the reference wavelength of each well was quantified using an MTP-310 absorbance microplate reader. Daunorubicin was used as a positive control. The results of cytotoxicity were recorded as growth inhibition percentages and as IC_50_ values [70,71].

#### 4.3.3. EGFR Inhibitory Assay

This assay was carried out according to the protocol for enzyme linked immunosorbent assay (ELISA) kits (Douset sandwich ELISA test, recombinant mouse EGFR) [72,73]. This assay employs the quantitative sandwich enzyme immunoassay technique. An antibody specific for EGFR has been pre-coated onto a microtiter plate. Standards or samples are pipetted into the wells and any EGFR present is bound by the immobilized antibody. After washing away any unbound substances, a biotin-conjugated antibody specific for EGFR is added to each well and incubated. Following a wash to remove unbound substances, streptavidin conjugated to Horseradish Peroxidase (HRP) are added to each microplate well and incubated. After washing away any unbound antibody–enzyme reagent, a substrate solution (TMB) is added to the wells and color develops in proportion to the amount of EGFR bound in the initial step. The color development is stopped by the addition of acid and the intensity of the color is measured at a wavelength of 450 nm ± 2 nm. The concentration of EGFR in the sample is then determined by comparing the O.D of samples to the standard curve.

#### 4.3.4. JNK-2 Inhibitory Assay

JNK-2 inhibitory assay was performed according to the protocol of enzyme-linked immunosorbent assay (ELISA). The assay was carried out using the JNK-2 kit (Simple Step ELISA, pT183/Y185, Abcam Company, Japan) [74,75] for the semi-quantitative measurement of JNK-2 protein in human cell lysate. The SimpleStep ELISA employs an affinity tag-labelled capture antibody and a reporter attached detector antibody to immunocapture the sample analyte in solution. This complete complex (capture antibody/analyte/detector antibody) is then immobilized via immunoaffinity of an anti-tag antibody coating the well. To perform the assay, samples or controls are added to the wells, followed by the antibody cocktail. After incubation, the wells are washed to remove unbound material. TMB substrate is added and during incubation is catalyzed by HRP, generating blue coloration. This reaction is then stopped by the addition of stop solution, completing any color change from blue to yellow. The signal is generated proportionally to the amount of bound analyte and the intensity is measured at 450 nm. Optionally, instead of the endpoint reading, the development of TMB can be recorded kinetically at 600 nm. An antibody cocktail can be prepared by combining an appropriate volume of the capture and detector antibodies immediately prior to assay. To make 3 mL of the antibody cocktail, combine 1.5 mL of capture antibody with 1.5 mL of detector antibody. Mix thoroughly and gently. Control lysate can be prepared from HEK293 cells, cultured in 10% FBS containing medium, then treated with 1 μg/mL anisomycin. After preparing all the reagents, samples, and control as instructed and following the previously published procedures [74,75], add 50 µL of sample or control to the well, add 50 µL of the antibody cocktail, then incubate at room temperature for 1 h on a plate shaker set to 400 rpm, aspirate and wash each well three times with 350 μL with wash buffer, add 100 μL TMB substrate to each well and incubate for 15 min, then add 100 μL stop solution and measure the absorbance at 450 nm.

#### 4.3.5. Apoptosis Analysis

For apoptosis induction measurement, seeding of Hela cells in 96-well plates (1 × 10^4^ cell/well) and DMEM medium was added and incubated for 24 h; at the second day, the cells were treated with the test compounds (at IC_50_, double and half of IC_50_) then incubated overnight; on the third day, each well was washed twice with 100 µL phosphate-buffered saline (PBS), add Trypsin 100 µL and incubate plates for 3 to 5 min at 37 degrees. Then, 200 µL of medium was added to separate cells, cells were centrifuged for 5 min, and supernatant was decanted. Then, it was washed twice with 100 µL PBS and 200 µL of buffer, 5 µL of PI& Annexin, and 200 µL of PBS was added for the test compounds. For control, 400 µL of PBS was added. Finally, the cell suspension was observed under the fluorescent microscope or transferred to a round bottom tube for flowcytometric analysis using (BD FACS melody).

#### 4.3.6. Cell Cycle Analysis

To cultured Hela cells in 96-well plates (1 × 10^4^ cell/well), DMEM medium, and the IC_50_ concentration of the test compounds were added, then incubated for 24 h after the addition of test compound. The supernatant was transferred to falcon tube for each plate and each well was washed twice with 3 mL PBS, Trypsin 1 mL was added, and plates were incubated for 3 to 5 min at 37 degrees, then 3 mL of the medium was added to separate cells, cells were centrifuged for 5 min, and supernatant was decanted. Then, it was washed again with 3 mL PBS, and 10 mL of medium was added. Cells were immediately fixed in ice cold 70% ethanol overnight at −20 °C. On the day of the analysis, cells were washed ×3 with PBS and re-suspended in propidium iodide (PI) for 15 min at room temperature, protected from light. Cell-cycle analysis was performed using flowcytometric analysis (BD FACS melody) and data obtained from cell cycle distribution was analyzed using FSC-W (Watson model) to estimate the percentage of cells in G1, S, and G2.

#### 4.3.7. Evaluation of Cytotoxicity against PC12 Cells

According to the standard water-soluble tetrazolium-8 (WST-8) assay, the cytotoxicity of compounds **8g** and **8i** against the PC12 cell line was evaluated, and daunorubicin was used as the reference drug by WST-8. The PC12 cells were maintained in a suspension culture, (Dulbecco’s modified eagle medium (DMEM), supplemented with 5% FBS (Fetal Bovine Serum) containing 1% of a penicillin-streptomycin (1:1) mixture. A 100 μL aliquot of cells (10,000 cells/mL) was added to a 96 well plate and incubated for 24 h at 37 °C in a humidified incubator containing 5% CO_2_ in the air. After 24 h, a 10 μL aliquot of test compound (concentrations varying in the range of 10–150 µM) was added to each of the 96 wells and incubated for 24 h. Then, a 10 μL WST-8 solution (mixture of WST-8 and 1-methoxy PMS) was added to each well and the incubation continued for 3 h. The visible absorbance at 450 nm and 630 nm as the reference wavelength of each well was quantified using an MTP-310 absorbance microplate reader. Daunorubicin was used as a positive control. The results of cytotoxicity were recorded as growth inhibition percentages and as IC_50_ values.

#### 4.3.8. Molecular Docking on EGFR and JNK-2

Docking simulation was performed by the Inter-coordinate Mechanics (ICM) using ICM-Pro 3.8 software (MolSoft L.L.C.) [76]. First, the 3D structures of the tested compounds and sorafenib (a reference multi-target kinase inhibitor) [77] were generated to perform well-suited docking. Then, the enzyme was prepared by adjusting the interface properties, including water molecules deletion, hydrogen atoms optimization, and formal charges refinement. In addition, enzyme relaxation was logged to run flexible docking. The ligands binding affinities were calculated by the Gaussian potential based on the ligand electrostatic potential and shape complementarity at the binding site [16,53]. In these studies, the template-docking method was used by selecting pre-defined binding pockets of the study receptors. The structural models of the tested compounds against human EGFR complexed with AZD9291 inhibitor (2.80 Å; PDB ID: 4ZAU) were used [78] and crystal structure of human JNK-2 complexed with an indazole inhibitor (2.14 Å; PDB ID: 3E7O) was used for c-Jun *N*-terminal kinase 2 (JNK-2) [15].

## Data Availability

All data generated or analysed during this study are included in this published article and its Appendix A Files.

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
