# Peer review of "Development and Assessment of 1,5–Diarylpyrazole/Oxime Hybrids Targeting EGFR and JNK–2 as Antiproliferative Agents: A Comprehensive Study through Synthesis, Molecular Docking, and Evaluation"

_molecules, 2023, doi:10.3390/molecules28186521_

Round 1
Reviewer 1 Report
The manuscript “Development and Assessment of 1,5-Diarylpyrazole/Oxime Hybrids Targeting EGFR and JNK-2 as Antiproliferative Agents: A Comprehensive Study Through Synthesis, Molecular Docking, and Evaluation” is devoted to the synthesis of a series of new pyrazole derivatives and evaluation of their biological activities. The work has a practical significance, since the results can be used for the development of new anticancer and neurotropic drugs. The supplementary information seems to be sufficient and comprehensive.
In my opinion, this manuscript suits to the scope of Molecules. After minor revision, it can be accepted.
However, I have some comments for the authors:
1. In introduction, a short explanations on the binding sites and their interactions with known inhibitors, developed compounds and structure of pharmacophore (fig. 3). are required.
2. scheme 1 – “conditions a” should describe both stages (condensation and acid hydrolysis).
3. In conclusions, protein-ligand interactions should be stated instead of binding free energies.
Reviewer 2 Report
The authors report the anticancer activity of a series of pyrazole oximes. The syntheses are routine but extensive biological evaluations are provided. Five cancer lines were used in this study and the cytotoxicity and antiproliferative activities were assessed. Emphasis is placed on the tyrosine kinase receptor and the results are further assessed in silica. The paper is not badly written but does have a number of minor grammatical errors. I am listing a few of these below (there are many further minor corrections needed). The only other issue that I have concerns the high resolution MS results. Many of the pyrazoles are known compounds and it is sufficient to compare the melting points with the literature values. For new compounds, HRMS results are provided. For this to be acceptable, the observed values must be within 0.003 amu of the calculated values. The values given for 6e and 6j are not even close to the required level of accuracy and these need to be rerun. Usually HRMS is given to 4 figures beyond the decimal point. The authors sometimes go with 3 figures, sometimes 4 and sometimes 5. The results need to be given consistently with 4 figures beyond the decimal point. The authors need to remember that HRMS is used to confirm the proposed molecular formulae and therefore accuracy and consistency is necessary to make this work.
Lines 73-75: while a number of these inhibitors .... due to its lack of specificity ...
Line 93: the authors might want to rethink the sentence "Oximes became fascinated in medicinal chemistry"
Line 106: is "psammaplin A analog" a natural product? The emphasis here is on "analog".
Line 110: Finally, oximes have
Lie 121: "small molecules" rather than "small compounds"
Line 147: diarylpyrazole
Line 161: were close to the calculated ones for all ...
Line 173: in the expected chemical shift range.
Line 178: were close to the calculated ones in ... Also, see lines 198 and 215.
For references 7 and 9, the reference numbers are duplicated.
As noted above, minor grammatical errors need to be addressed.
